# BioDataLab: Towards Generalist Agents for Real-world Biological Data Engineering

## Abstract

Automating the intricate process of dataset construction, known as Biological Data Engineering (BDE), is a grand challenge for autonomous AI agents and a critical bottleneck in scientific discovery. While Large Language Models (LLMs) show promise, their application is hampered by the absence of a rigorous benchmark to guide and evaluate agent development in this domain. To address this gap, we introduce BioDataLab, the first comprehensive benchmark designed to operationalize BDE and drive progress in scientific automation. BioDataLab features 114 realistic tasks curated from 150 peer-reviewed biological publications. It systematically tackles core scientific challenges by: (1) managing procedural ambiguity with clear goals but open-ended execution paths; (2) establishing intermediate ground truth by manually replicating each task with tractable data; and (3) enabling complex, multi-modal evaluation through custom, domain-aware evaluators for specialized scientific data formats beyond simple string matching. We conduct an extensive evaluation of state-of-the-art agents powered by models such as GPT-4.1, Claude 4, and Gemini 2.5. Our results reveal that while these models exhibit nascent capabilities, their overall success rates are modest, exposing a significant performance gap. We identify critical and recurrent failure modes, including struggles with multi-step tool chaining, hallucination of tool parameters, inability to parse scientific file formats, and a lack of long-horizon reasoning. These findings not only validate the challenging nature of BDE but also provide a granular, empirical roadmap for the community to develop more robust and reliable scientific agents.

## 1 Introduction

Data-driven approaches have revolutionized biological sciences, enabling breakthroughs across genomics Han et al. (2025), structural biology Gong et al. (2024), and drug discovery Ünlü et al. (2025). However, their success fundamentally depends on the availability of high-quality, well-structured datasets Wang et al. (2005). Building such datasets is far from trivial Gao et al. (2022): it requires a complex, labor-intensive process we term **Biological Data Engineering (BDE)**. BDE encompasses tasks such as querying heterogeneous databases to collect source data, running domain-specific bioinformatics tools to extract desired features, finishing custom data processing, and reasoning to integrate and validate results A.4. This demanding process has emerged as a critical bottleneck in the scientific discovery pipeline Rigden & Fernández (2024). Its importance is underscored by top-tier journals like *Nucleic Acids Research* (NAR), which publish hundreds of papers annually A.1 on newly curated datasets (e.g., PDBbind Wang et al. (2005), ChEMBL Gaulton et al. (2012)), highlighting the central role of BDE in advancing biological research.

Despite its critical role, BDE remains a predominantly manual endeavor, consuming vast amounts of researcher time and effort Cao et al. (2022). The rise of Large Language Model (LLM) -based autonomous agents, which are proven effective in automating complex tasks such as web navigation Deng et al. (2023) and software engineering Qiu et al. (2025), offers a promising path to alleviate this bottleneck. However, there is no systematic framework that decomposes BDE into measurable, automatable tasks. Even more limiting is the absence of a dedicated, high-quality benchmark to guide the design, development, and evaluation of agents tailored for BDE.

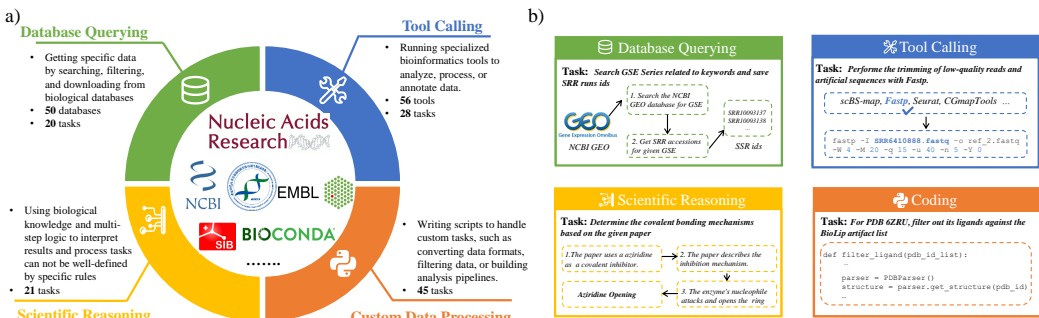

Figure 1: Overview of real-world BDE benchmark BIODATALAB. (a) The four categories BDE tasks defined in BIODATALAB. (b) Illustrative examples of different categories BDE tasks.

This paper addresses this gap by introducing BIODATALAB, the first comprehensive benchmark for real-world biological data engineering. Our central thesis is that a carefully constructed benchmark can operationalize the definition of BDE, transforming it from a vaguely described process into a set of concrete, measurable tasks. Building such a benchmark required us to confront three fundamental challenges inherent to the scientific process. First, procedural ambiguity: research papers Gao et al. (2022) often specify goals (e.g., "retrieve relevant protein sequences") but omit the precise procedures, such as the exact database queries or tool parameters used. Second, lack of intermediate ground truth: publications Cao et al. (2022) typically report only the final dataset, leaving the intermediate steps of the data engineering pipeline unverified. Third, multi-modal and complex evaluation: BDE tasks Decker et al. (2022) generate outputs ranging from structured text to specialized scientific files (e.g., FASTA, PDB), necessitating domain-specific validation logic that goes far beyond simple string matching ( S1).

We designed BIODATALAB to systematically address these challenges as shown in Figure 1. To tackle procedural ambiguity, we clearly define each task's intent while leaving the specific execution details for the agent to determine. To provide reliable ground truth and enable rapid evaluation, we manually replicated all 114 tasks—sourced from 150 NAR papers—using representative, computationally feasible data samples (e.g., the pumpkin genome instead of the human genome), generating verifiable inputs and outputs. For diverse evaluation, we developed custom evaluation functions for each task, capable of parsing and validating complex scientific data formats. All tasks within BIODATALAB are manually created and checked by human experts.

To demonstrate BIODATALAB 's utility and establish strong baselines, we conducted an extensive evaluation of several state-of-the-art LLMs, including GPT OpenAI (2024), Claude Anthropic (2025), and Gemini Google (2025), integrated within a dedicated agent framework. The results are revealing: while these powerful models show early promise, their overall success rates remain modest, highlighting both the intrinsic difficulty of BDE and a substantial capabilities gap in current agent technology. Our analysis uncovers several critical and recurring failure modes: agents often struggle with multi-step tool chaining, hallucinate incorrect parameters for bioinformatics tools, mishandle specialized scientific data formats, and lack the long-horizon reasoning required to complete complex workflows. These findings not only validate the challenging nature of BIODATALAB, but also provide a detailed empirical roadmap for the community to develop more capable and reliable scientific agents.

In summary, our main contributions are:

• We formalize and operationalize Biological Data Engineering (BDE) as a critical challenge for AI agents, proposing a taxonomy of four core task categories that systematically dissect the domain.

• We introduce **BIODATALAB**, a diverse, multi-modal benchmark featuring 114 real-world tasks derived from scientific literature, complete with manually prepared ground-truth data and custom evaluators to enable rigorous and reproducible agent evaluation.

- We provide a comprehensive analysis of leading LLMs on BIODATALAB, establishing strong baselines and identifying key failure modes, thereby offering a clear roadmap for the development of capable BDE agents.

## 2 RELATED WORK

**LLM Agents for Data Science.** LLM-based agents are increasingly used to automate end-to-end data science workflows, extending beyond code generation for isolated tasks to full pipeline orchestration. Systems such as DatawiseAgent You et al. (2025) employ notebook-centric planning and self-debugging to execute multi-step analyses, while AutoMind Ou et al. (2025) combines expert knowledge bases with adaptive search for context-aware solutions. Case-based agents like DS-Agent Guo et al. (2024) further exploit prior expert workflows to guide model selection and experiment design. Data Interpreter Hong et al. (2024) models the data science workflow as a hierarchical graph of tasks and actions, enabling dynamic planning and iterative self-correction based on execution outcomes. Recent surveys Wang et al. (2025) highlight these developments and point to open challenges, including scalability, robustness, and explainability, as the field moves from task-specific assistants toward autonomous agents capable of executing complex data science processes with minimal human oversight.

**LLM Agents for Bioinformatics.** The application of Large Language Models (LLMs) and autonomous agents in the biological sciences is a rapidly expanding frontier. Early efforts focused on using LLMs for specific, well-defined tasks such as mining scientific literature Zheng et al. (2023); Kang & Kim (2024). More recently, agent-based systems have emerged that can interact with tools and execute complex workflows. For instance, ChemCrow was developed to automate tasks in chemical synthesis and drug design by integrating multiple expert-designed tools Bran et al. (2023). Similarly, AutoBA Zhou et al. (2024a) focuses on automating multi-omics data analysis pipelines. Closer to our work are generalist biological agents such as Biomni Huang et al. (2025) and STELLA Jin et al. (2025), which aim to tackle a wide array of bioinformatics tasks by leveraging LLMs to interpret user requests and invoke appropriate tools.

**Benchmark for LLM Agents.** With the proliferation of agent research, rigorous and standardized evaluation has become paramount. Several comprehensive benchmarks have been proposed to assess agent capabilities. General-purpose benchmarks like AgentBench Liu et al. (2023) and GAIA Mialon et al. (2023) evaluate agents across a wide spectrum of tasks, testing their reasoning, decision-making, and tool-use abilities in diverse contexts. Other benchmarks focus on specific capabilities; for example, ToolBench Qin et al. (2023) and API-Bank Li et al. (2023) are designed to exclusively measure an agent's proficiency in using external tools and APIs. Domain-specific benchmarks have also been crucial, with SWE-bench Jimenez et al. (2023) and WebArena Zhou et al. (2024b) serving as de facto standards for evaluating agents in software engineering and web navigation, respectively.

## 3 THE BIODATALAB BENCHMARK

We introduce BIODATALAB, a comprehensive benchmark designed to evaluate agent capabilities in Biological Data Engineering (BDE). It is meticulously curated from 150 peer-reviewed publications and comprises 114 real-world tasks. These tasks involve interactions with 50 foundational public databases and the use of 56 domain-specific bioinformatics tools. In the following sections, we formally define BDE tasks, describe our principled benchmark construction pipeline, and present a detailed statistical analysis of the resulting dataset.

### 3.1 TASK TAXONOMY AND FORMALISM

The primary objective of BIODATALAB is to assess an agent's ability to perform BDE in realistic scenarios. To this end, we first establish a structured taxonomy of BDE activities. As illustrated in Figure 1, we deconstruct the BDE process into four core task categories:

- **Database Querying**: Retrieving or downloading specific data from online APIs or local database files. This category tests an agent's ability to comprehend database schemas, formulate correct queries, and handle data retrieval protocols.

Figure 2: The construction process of BIODATALAB. (a) Source of BIODATALAB are high-quality paper published in the top-tier journal Nucleic Acids Research (NAR) . (b) Human-in-the-loop curation pipeline. (c) For each selected BDE steps, we manually convert it into a standard benchmark tasks.

- **Tool Using**: Selecting and executing domain-specific bioinformatics tools to perform data analysis or feature annotation. This assesses an agent's capacity for correct tool selection, parameterization, and chaining in multi-step analytical pipelines.
- **Custom Data Processing**: Generating custom scripts for tasks such as data parsing, format conversion, filtering, and bespoke analysis. This evaluates an agent's ability to translate scientific requirements into functional and correct code.
- **Scientific Reasoning**: Performing other reasoning tasks that require deep domain knowledge, such as summarizing biological mechanisms or validating data integrity based on scientific principles. This directly probes an agent's understanding of the underlying biology.

Formally, we define each BDE task as a 5-tuple: $\text{Task}(E, T, I, F, M)$. Here, $E$ is the interactive environment, comprising a "tool ocean" and a "database ocean" that contain all necessary software and data resources. $T$ is the task instruction, a natural language description of the goal. $I$ is the set of initial input files required for the task. $F$ is a custom evaluation function that programmatically verifies the correctness of the final output. Finally, $M$ is the foundational LLM that powers the agent being evaluated.

## 3.2 BENCHMARK CURATION PIPELINE

Based on above definition, we describe the principled, multi-stage pipeline we developed to construct a diverse, concise, and realistic benchmark from the vast corpus of Nucleic Acids Research (NAR) database publications.

### 3.2.1 SOURCE MATERIAL SELECTION

The NAR database issue is a premier venue, publishing over 100 papers on new datasets annually. To ensure our benchmark reflects contemporary scientific practice, we established the following selection criteria: (1) **Timeliness**: We focused on papers published between 2021 and 2024 to avoid deprecated tools or obsolete data sources; (2) **Disciplinary Diversity**: To ensure broad coverage, we sampled papers from all 15 major categories defined by NAR, deliberately covering diverse sub-fields of biology; (3) **Quality and Impact**: We selected only those databases that were featured in both the annual NAR database issue and are listed in the NAR Online Molecular Biology Database Collection. This dual-inclusion criterion ensures that we select papers describing well-established and impactful datasets. This rigorous filtering process yielded a high-quality set of 150 source publications.

### 3.2.2 TASK SOURCING AND CURATION

A naive extraction of every data engineering step from the 150 papers would result in a benchmark that is redundant, imbalanced, and computationally intractable. For instance, a few common tools

would dominate the tasks, while many simpler steps would offer little evaluative value. Therefore, we designed a principled curation strategy to distill the most representative and challenging tasks.

First, we manually decomposed the full data construction workflow described in each paper into atomic BDE steps. This process yielded an initial pool of 1,489 raw steps. A preliminary analysis revealed a significant imbalance: Tool Using (687), Custom Data Processing (379), Database Querying (218), and Scientific Reasoning (110). Besides, we observed high redundancy of tool using and database querying steps where 10-20% of tools and databases accounted for 70-80% of the usage instances.

To construct a balanced and effective benchmark, we applied a selection strategy guided by three core principles: (1) **Diversity**: The final task set must comprehensively represent the variety of BDE challenges encountered in practice; (2) **Conciseness**: The benchmark should be as small as possible while maintaining representativeness, enabling efficient evaluation; (3) **Realistic Challenge**: Tasks should be non-trivial for current agents but reflect real-world procedures, avoiding artificially contrived problems.

Applying these principles, we curated the final set of tasks. For 'Scientific Reasoning', we retained nearly all unique steps due to their inherent diversity and smaller number. For 'Tool Using' and 'Database Querying', we ensured the inclusion of the most frequently used resources while sampling a diverse set of less common ones. For 'Custom Data Processing', we categorized tasks along two axes—data structures manipulated and biological objectives—to ensure comprehensive coverage of typical scripting needs. This curation process resulted in our final set of 114 benchmark tasks, with a more balanced distribution: Tool Using (28), Custom Data Processing (45), Database Querying (20), and Scientific Reasoning (21) as shown in Figure. 1.

### 3.2.3 FROM RAW STEPS TO EXECUTABLE TASKS

Transforming the curated steps into high-quality, executable benchmark tasks required surmounting three fundamental challenges, mirroring the issues we highlighted in the introduction.

**Procedural Ambiguity.** Scientific papers often describe procedures with ambiguous language. We distinguish between intent ambiguity (the scientific goal is unclear) and procedural ambiguity (the goal is clear, but the exact execution steps are omitted). BIODATALAB is designed to test an agent's ability to resolve procedural ambiguity, not to guess scientific intent. For example, a task like "retrieve gut microbiome data from healthy animals" is intentionally ambiguous. In contrast, "retrieve all SRR accessions for GSExxxxx from NCBI GEO" is procedurally ambiguous—the goal is precise, but the agent must determine how to achieve it.

To address this, we systematically refined each task description to ensure the intent is unequivocal. For instance, database queries are specified with explicit keywords and timeframes, and tool use tasks are framed such that an agent must infer appropriate parameters from the context (e.g., by using default settings or reasoning based on the data), rather than being given them explicitly. This ensures that our evaluation is robust and grounded in solving well-defined problems.

**Lack of Intermediate Ground Truth.** Publications typically only provide the final dataset, leaving no ground truth for the intermediate steps of the data engineering pipeline. To solve this, we generated verifiable inputs and outputs for every task. This involved two key actions. First, we instantiated each task with representative yet computationally tractable data samples. For example, for a task involving mapping sequencing reads to a genome, instead of using the entire human genome (hundreds of gigabytes), we substituted it with the pumpkin genome (about 100MB). This retains the task's essential complexity while making evaluation feasible and rapid. Tasks for which no such tractable sample could be found were discarded. Second, our team manually executed each of the 354 tasks to generate gold-standard outputs, providing reliable ground truth for evaluation.

**Complex and Multi-modal Evaluation.** Unlike typical benchmarks that rely on simple string matching or multiple-choice answers, BDE tasks produce a diverse array of outputs, including structured text, tables, and specialized scientific file formats (e.g., FASTA, PDB, VCF). To enable rigorous validation, we engineered a custom evaluation function for each task. These evaluators are designed to parse complex scientific data formats, compare key information content, and apply

Table 1: Comparison with existing related benchmarks.

| Benchmark | Real World | Domain Knowledge | Multi-step | Code | Objective Evals | Multi-Modal | #Tasks |
|---|---|---|---|---|---|---|---|
| ARCADE Yin et al. (2023) | ✗ | ✗ | ✓ | ✓ | ✓ | ✗ | 10,082 |
| DABENCH Hu et al. (2024) | ✓ | ✗ | ✓ | ✓ | ✓ | ✗ | 257 |
| DA-Code Huang et al. (2024) | ✓ | ✗ | ✓ | ✓ | ✓ | ✗ | 500 |
| DS-1000 Lai et al. (2023) | ✓ | ✗ | ✗ | ✓ | ✓ | ✗ | 1,000 |
| SPIDER2V Cao et al. (2024) | ✓ | ✓ | ✗ | ✓ | ✓ | ✗ | 494 |
| DSEVAL Zhang et al. (2024) | ✗ | ✗ | ✓ | ✓ | ✓ | ✗ | 825 |
| DSBENCH Jing et al. (2025) | ✗ | ✗ | ✓ | ✓ | ✗ | ✓ | 540 |
| DABSTEP Egg et al. (2025) | ✓ | ✓ | ✓ | ✓ | ✓ | ✗ | 450 |
| COTA Li et al. (2025) | ✓ | ✗ | ✓ | ✓ | ✓ | ✓ | 1013 |
| BAISBENCH Luo et al. (2025) | ✓ | ✓ | ✓ | ✓ | ✓ | ✗ | 229 |
| BIOCODER Tang et al. (2024) | ✓ | ✓ | ✗ | ✓ | ✓ | ✗ | 3189 |
| BIXBENCH Mitchener et al. (2025) | ✓ | ✓ | ✓ | ✓ | ✗ | ✗ | 296 |
| GENOME-BENCH Yin et al. (2025) | ✓ | ✓ | ✗ | ✗ | ✗ | ✓ | 3332 |
| **BIODATALAB (Ours)** | ✓ | ✓ | ✓ | ✓ | ✓ | ✓ | 114 |

domain-specific logic to verify correctness far beyond superficial textual similarity. This ensures that our evaluation is both automated and scientifically meaningful. Details of benchmark construction can be found in Appendix B.1

## 3.3 BENCHMARK STATISTICS AND FEATURES

BIODATALAB is designed to be a comprehensive and challenging testbed for scientific agents, uniquely positioned at the intersection of data science, bioinformatics, and multi-step reasoning. To contextualize its contribution, we compare it with existing benchmarks in data science and bioinformatics in Table 1. While prior work has made significant strides, BIO-DATALAB is the first to systematically integrate real-world procedural ambiguity, deep domain knowledge, complex multi-modal data formats, and rigorous, objective evaluation within a single framework.

Table 2: Basic statistics on of BIODATALAB.

| Dataset Component | Count |
|---|---|
| # of Tasks | 114 |
| # of Evaluation Functions | 22 |
| # of Tools Included | 56 |
| # of Databases Included | 50 |

For instance, benchmarks like DS-1000 and BIOCODER excel at evaluating single-step coding proficiency but do not capture the multi-step, tool-chaining nature of real scientific workflows. Conversely, benchmarks such as DABENCH and BIXBENCH address multi-step processes but often lack the deep, specialized domain knowledge required for tasks in BIODATALAB, or they rely on human-in-the-loop evaluation. BIODATALAB synthesizes these dimensions: its tasks are not only multi-step and grounded in real-world publications but also demand an understanding of biological concepts to correctly select and parameterize tools, and they produce outputs that are objectively and automatically verifiable.

The key features that distinguish BIODATALAB are:

**Fidelity to Real-World Scientific Workflows.** The benchmark's core strength lies in its authenticity. All 114 tasks are directly derived from the data engineering pipelines described in 150 recent NAR publications. This grounding ensures that agents are evaluated on problems that biologists actually face. The tasks require interaction with a diverse ecosystem of 50 distinct public databases and the correct application of 56 unique bioinformatics tools, reflecting the true heterogeneity of the BDE landscape.

**Rich Multi-modality and Complex Data Formats.** A defining feature of BDE is the diversity of data representations, a challenge not fully captured by existing benchmarks. As illustrated in Table S1, tasks in BIODATALAB require agents to parse, manipulate, and generate a wide array of specialized scientific file formats. These are not limited to simple text or tabular data; they include

structured formats for DNA/protein sequences (FASTA, GenBank), 3D molecular structures (PDB, SDF), genomic variations (VCF), and sequence alignments (SAM/BAM). This multi-modality tests an agent's ability to handle complex data structures and move beyond generic text-based reasoning, a critical capability for any true scientific agent.

**Domain-Aware, Objective Evaluation.** Evaluating scientific tasks demands more than simple string matching. An agent might generate a file with a correct-looking filename or header, but with scientifically invalid content. To address this, we developed a suite of 22 custom evaluation functions. These evaluators act as semantic parsers, programmatically validating the scientific correctness of the output. This ensures our evaluation is both automated and scientifically rigorous, providing a reliable measure of an agent's true capabilities.

Collectively, these features make BIODATALAB a formidable challenge that systematically probes the limitations of current agents. The average task instruction length is 78 tokens, providing a concise yet unambiguous goal, while requiring the agent to generate complex, multi-step solutions to succeed. The benchmark thus serves not only as an evaluation tool but also as a clear roadmap for developing the next generation of generalist scientific agents. Details of benchmark statistic and examples can be found in Appendix B.2 and B.3

## 4 EXPERIMENTAL FRAMEWORK AND EVALUATION

### 4.1 AGENT FRAMEWORK

To conduct a rigorous evaluation on BIODATALAB, we require a capable agent framework that can translate an LLM's reasoning into executable actions. We employ Biomni Huang et al. (2025), a general-purpose agent designed for scientific workflows. Its key feature is the use of code as a universal action interface. Instead of relying on predefined function calls, the agent generates and executes code (e.g., Python scripts, shell commands) to orchestrate its workflow. This code-centric approach provides the flexibility needed to chain tools, perform custom data manipulations, and implement complex logic, closely mirroring how human experts perform BDE. The agent architecture also incorporates retrieval-augmented planning to select relevant tools and an iterative refinement loop to adapt its plan based on execution outcomes. For this work, Biomni serves as a standardized testbed; our focus is on evaluating the reasoning capabilities of the underlying LLMs, not the agent framework itself. Further details of the agent can be found in Appendix C

### 4.2 MODELS AND EVALUATION

We evaluated a suite of state-of-the-art LLMs, using them as the reasoning engine within the Biomni agent framework.

- **Proprietary Models:** We evaluated leading closed-source models, including OpenAI's GPT 4.1, Anthropic's Claude 4, and Google's Gemini 2.5. These models represent the current frontier of LLM capabilities.
- **Open-Source Models:** We also benchmarked top-performing open-source models, including Qwen 3, and DeepSeek V3.1.

**Evaluation Metric.** Our primary metric is **Success Rate (SR)**, defined as the percentage of tasks for which the agent produces a final output that passes the custom, domain-aware evaluation function. Each task is considered a single, independent trial.

## 5 EXPERIMENTAL RESULTS

### 5.1 MAIN RESULTS

The performance of all evaluated models on BIODATALAB is presented in Table 3 and Figure. 3. Our analysis yields several key findings:

**1. Significant Capabilities Gap and the Intrinsic Difficulty of BDE:** Across all models, even the most advanced proprietary LLMs like GPT-4.1, the overall success rates remain strikingly low,

Table 3: Performance of different LLM on BIODATALAB. Scores reflect Success Rate (SR) on different categories BDE tasks. The best results are **bolded**.

| Name | Tool Using (28) | Custom Data Processing (45) | Database Querying (20) | Scientific Reasoning (21) | Total (114) |
|------|------|------|------|------|------|
| *Closed-source Models* | | | | | |
| GPT-4.1 | **42.9** | **51.1** | 25.0 | 19.0 | **38.5** |
| GPT-4.1-MINI | 35.7 | 44.4 | 30.0 | 4.8 | 32.4 |
| GEMINI-2.5-PRO | 35.7 | 33.3 | **45.0** | 23.8 | 34.2 |
| GEMINI-2.5-FLASH | 21.4 | 33.3 | 10.0 | 4.7 | 21.1 |
| CLAUDE-SONNET-4 | 25.0 | 46.7 | 40.0 | 9.5 | 33.2 |
| CLAUDE-SONNET-3.7 | 21.4 | 40.0 | 30.0 | 19.0 | 29.8 |
| *Open-source Models* | | | | | |
| DEEPSEEK-V3.1-REASONER | 32.1 | 44.4 | 25.0 | 23.8 | 34.2 |
| DEEPSEEK-V3.1-CHAT | 25.0 | 42.2 | 30.0 | 19.0 | 32.0 |
| QWEN3-CODER-480B-A35B-INSTRUCT | 25.0 | 40.0 | 30.0 | **28.6** | 32.5 |

with the top performer achieving only 38.5%. This modest performance unequivocally validates the challenging nature of Biological Data Engineering and underscores a significant capabilities gap in current autonomous agents. The results suggest that BDE tasks, characterized by their procedural ambiguity, multi-step nature, specialized tool usage, and complex data formats, push the boundaries of what even state-of-the-art LLMs can handle effectively in an agentic setup. This highlights BIO-DATALAB's effectiveness in revealing the current limitations of LLM agents in scientific domains.

**2. Varied Performance Across Task Categories with Custom Data Processing as a Relative Strength:** While overall performance is low, there are notable differences across the four BDE task categories. 'Custom Data Processing' consistently shows the highest success rates for most models, with GPT-4.1 achieving 51.1%. This suggests that current LLMs, particularly powerful ones, are relatively adept at generating custom scripts for data manipulation, parsing, and filtering when the problem is well-defined. In contrast, 'Tool Using' and 'Database Querying' present a more substantial hurdle, and 'Scientific Reasoning' proves to be the most challenging category for most models, with many scoring below 20%. This indicates a stronger proficiency in general-purpose coding compared to the intricate knowledge required for precise tool parameterization, complex database interactions, or abstract biological reasoning.

**3. Open-Source Models Approach Proprietary Baselines, but GPT-4.1 Leads Overall:** The performance gap between proprietary models (GPT-4.1, Claude-4, Gemini-2.5) and leading open-source models (DeepSeek-v3.1, Qwen3) is present but not insurmountable, with DeepSeek-v3.1-reasoner and Qwen3-coder achieving total success rates of 34.2% and 32.5% respectively, quite close to some proprietary offerings like Gemini-2.5-pro (34.2%) and Claude-sonnet-4 (33.2%). This suggests that advancements in open-source LLMs are closing the gap, particularly in tasks requiring more general coding or reasoning capabilities. However, GPT-4.1 remains the overall strongest performer, especially in 'Tool Using' and 'Custom Data Processing', indicating its superior ability to orchestrate complex actions and generate precise code within the scientific context. The competitive performance of open-source models offers promising avenues for future research and development in scientific agents.

**4. Model-Specific Patterns in Trajectory Length:** GPT-4.1 shows a comparatively wider spread of successful (green) tasks across various trajectory lengths, with a noticeable presence in the 20-40 turn range, indicating its ability to tackle more intricate, multi-step problems. Its failures, while still numerous at shorter lengths, also extend to longer trajectories, pointing to challenges in very complex or subtle aspects of BDE. Claude-sonnet-4 and Gemini-2.5-pro display similar patterns to GPT-4.1 in that successful tasks often require longer trajectories, but with a generally higher density of failures across all lengths, especially at the shorter end. This aligns with their slightly lower overall success rates. Open-source models (DeepSeek-v3.1-reasoner, Qwen3-coder), while showing promising overall performance, still exhibit a stronger concentration of failures at shorter trajectory lengths compared to GPT-4.1, suggesting that their initial planning and robust execution in the early stages of a task might still lag behind the top proprietary models. However, they also manage to achieve success on tasks requiring moderate trajectory lengths, showcasing their growing capabilities.

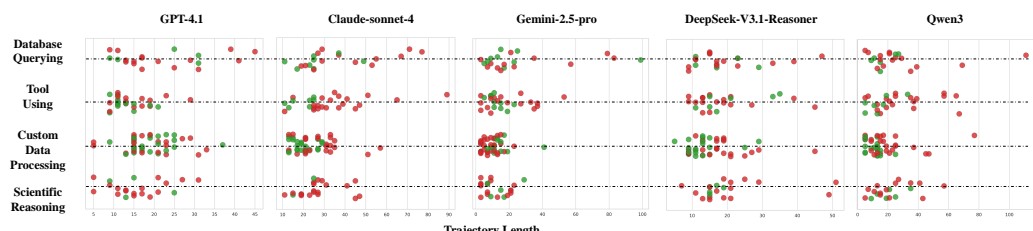

Figure 3: The distribution of length of interaction trajectory over different models. The red points are failed samples and the green are success.

## 5.2 ANALYSIS OF FAILURE MODES

To understand the underlying reasons for the modest success rates, we performed a detailed qualitative analysis of failed trajectories and categorized recurring failure modes. Figure 4 visually summarizes the distribution of these failure reasons across the evaluated models.

The detailed analysis of failure modes, as depicted in the figure, reveals that brittle tool chaining and error recovery (A) is the most pervasive issue, affecting all models significantly, particularly Claude-sonnet-4 and GPT-4.1, highlighting a fundamental difficulty in orchestrating multistep scientific workflows. Hallucination of tool parameters and file paths (B) is also a considerable challenge, especially for Gemini-2.5-pro, indicating struggles with precise contextual

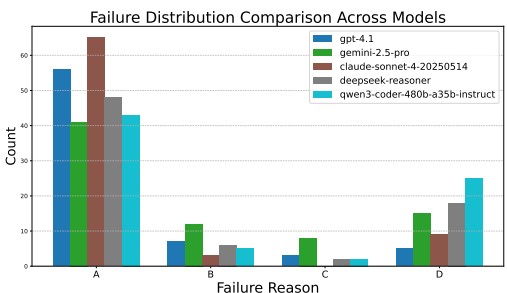

Figure 4: The failure distribution over different models.

grounding of tool knowledge. While less frequent, the inability to parse and manipulate specialized scientific file formats (C) contributes to failures, again more so for Gemini-2.5-pro, underscoring a weakness in handling complex data structures. Finally, a lack of long-horizon reasoning (D) poses a major obstacle, predominantly for open-source models like Qwen3-coder and DeepSeek-v3.1-reasoner, suggesting a need for enhanced capabilities in maintaining coherent plans over extended task sequences. Collectively, these insights pinpoint critical areas for improving agent robustness and scientific domain adaptation.

## 6 CONCLUSION

In this work, we introduced BIODATALAB, the first comprehensive benchmark for Biological Data Engineering. By systematically curating 114 real-world tasks from scientific literature and developing a rigorous, domain-aware evaluation framework, we have operationalized BDE as a grand challenge for AI agents. Our extensive evaluation of state-of-the-art LLMs reveals that while these models are beginning to show promise, their capabilities fall short of the demands of real-world scientific workflows. The identified failure modes—brittle tool chaining, parameter hallucination, poor handling of scientific data formats, and limited long-horizon reasoning—provide a clear and actionable roadmap for the research community. We release BIODATALAB to the public as a catalyst for innovation, aiming to guide the development of the next generation of more capable, reliable, and ultimately more useful scientific AI agents.

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

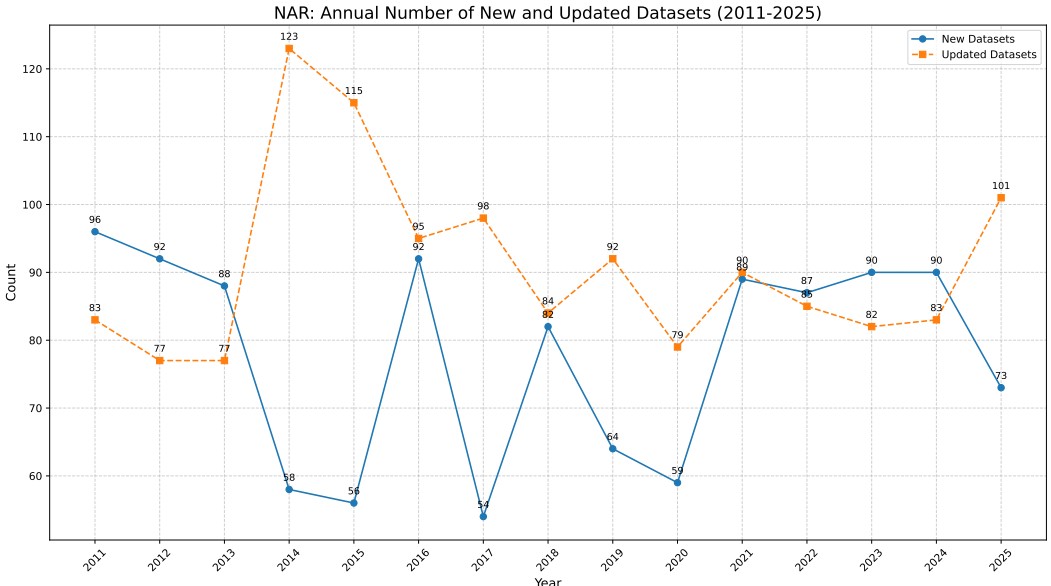

Figure S1: The annual number of new and updated datasets featured in the Nucleic Acids Research (NAR) from 2011 to 2025.

## A DETAILS OF BIOLOGICAL DATA ENGINEERING

### A.1 DATABASE ISSUE OF NUCLEIC ACIDS RESEARCH

Figure S1 illustrates the annual number of new and updated datasets featured in the Nucleic Acids Research (NAR) journal from 2011 to 2025, revealing a dynamic relationship between database creation and maintenance. While both categories show sustained activity, a key observation is the inverse trend during 2014-2015, where a significant drop in new dataset submissions coincided with a dramatic peak in updates, suggesting a community-wide focus on improving existing resources. Outside of this period, the generally comparable numbers for both new and updated datasets reflect a mature bioinformatics ecosystem that maintains a healthy balance between innovation and the long-term sustainability of its core data infrastructure.

### A.2 CATEGORIES OF DATABASE PAPERS

Figure S2 presents a treemap that visually represents the categorical distribution of publications in the NAR database issue. The visualization immediately highlights the dominance of foundational 'omics' fields, with 'Genomics', 'Protein sequence' and 'Nucleotide Sequence' constituting the largest shares of publications. Following closely are areas directly related to human health and systems biology, such as 'Human Genes and Diseases' and 'Metabolic & Signaling Pathways' indicating significant community investment in these translational and integrative domains. Conversely, the treemap also illustrates the breadth of the field by including numerous smaller, more specialized categories like 'Organelle' and 'Cell biology', representing fewer publications, underscore the diverse applications of molecular biology databases. The use of area and corresponding font size effectively creates a visual hierarchy, allowing for a rapid assessment of the primary focus areas within the bioinformatics resource landscape as captured by NAR.

### A.3 SCIENTIFIC DATA FORMAT OF DATABASE PAPERS

Table S1 provides a systematic framework for classifying bioinformatics data files by decoupling their physical Data Structure (rows) from their Biological Semantics (columns). The vertical axis, representing the data structure, addresses the computational challenge of data access and parsing, ranging from simple, human-readable tabular and sequential formats to complex, indexed binary or

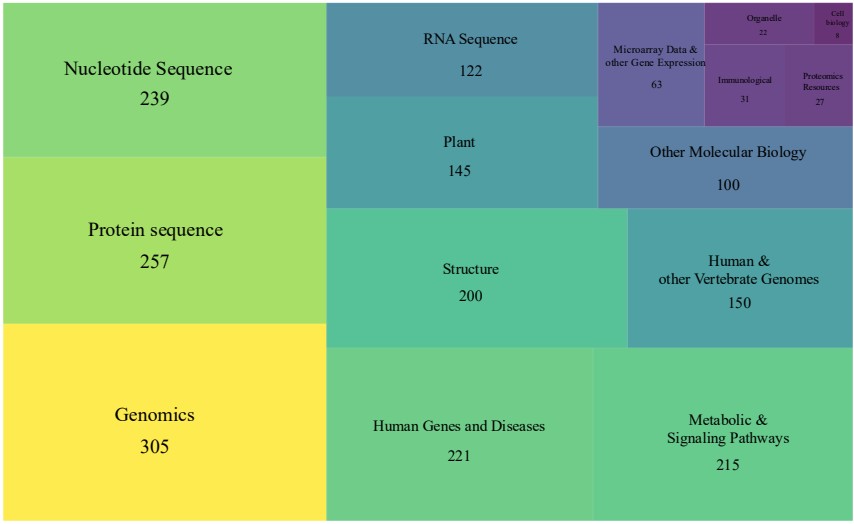

Figure S2: A treemap visualizing the breakdown of NAR database publications by category.

Table S1: Classification of Bioinformatics Data Structures and Formats.

| | A. Sequence & Variation | B. Genomic Coordinates & Annotation | C. Quantitative Measurement | D. Relationships & Networks | E. Chemical & Structural | F. Statistical & Analytical Results | G. Metadata |
|---|---|---|---|---|---|---|---|
| **Tabular** | VCF | BED / GFF / GTF Gene Lists | Expression Matrix | ✗ | ✗ | Enrichment Results (P-value, FDR, LogFC) | Metadata Table |
| **Sequential** | FASTA / FASTQ | ✗ | ✗ | ✗ | ✗ | ✗ | ✗ |
| **Hierarchical / Object-based** | ✗ | Genome Annotations (JSON, XML) | Single-cell Objects (H5AD, RDS) HDF5 | Regulatory Networks (JSON, XML) | PDB | Results within analysis objects | Metadata within single-cell objects |
| **Binary** | BAM | ✗ | BigWig .raw (Mass Spec) | ✗ | ✗ | ✗ | ✗ |
| **String / Text-based** | ✗ | ✗ | ✗ | Phylogenetic Tree (Newick) | SMILES InChIKey | ✗ | ✗ |

hierarchical object-based files that require specialized libraries. The horizontal axis, representing the biological semantics, addresses the domain-specific challenge of interpretation, spanning concepts from fundamental sequence information and genomic coordinates to high-level quantitative measurements and network relationships. The key insight from this framework is the non-exclusive relationship between these two dimensions; for example, a single semantic category like 'Quantitative Measurement' can be represented across multiple structures (e.g., a tabular expression matrix, a hierarchical H5AD object, or a binary BigWig file). This classification is therefore crucial for understanding the dual nature of bioinformatics tasks, separating the foundational computational capability to parse a file from the advanced, context-aware ability to interpret and analyze its biological meaning.

## A.4 TYPICAL BDE EXAMPLES

**COVPDB.** Gao et al. (2022) The data engineering workflow for constructing the COVPDB, as illustrated in the Figure S3, begins with Database Querying of the PDB and linking to external resources like ChEMBL and UniProt to aggregate an initial dataset. This comprehensive collection then undergoes Custom Data Processing, where Scientific Reasoning is applied to filter and retain only high-resolution, biologically relevant covalent complexes, while discarding crystallization artifacts and other non-specific entries. Finally, the curated dataset is extensively enriched through a hybrid approach: automated Tool Using calculates specific physicochemical properties such as Molecular Weight, SASA, and pKa, while meticulous Manual curation is employed to annotate complex features like the covalent bonding mechanism and binding affinity, ensuring a high-quality and richly detailed final resource for downstream applications.

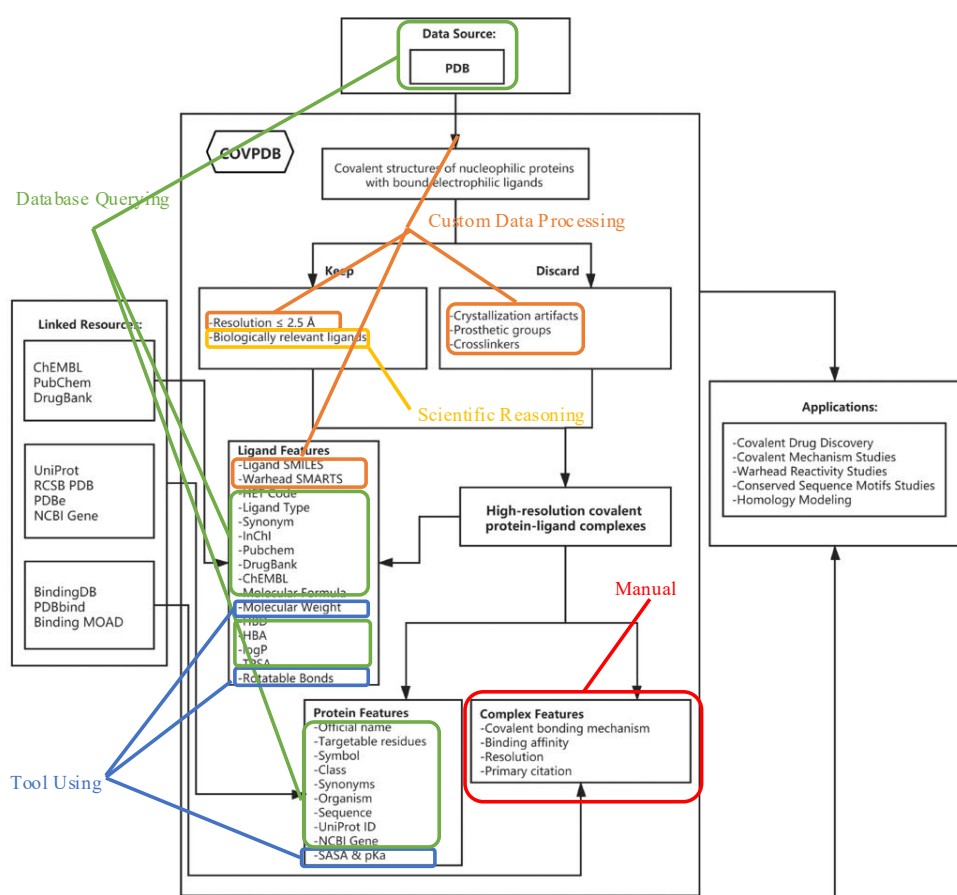

Figure S3: The data engineering workflow for the construction of COVPDB

**webTWAS.** Cao et al. (2022) As in Figure S4, the illustrated workflow commences with Database Querying to aggregate GWAS summary statistics from diverse sources, including the UKBB cohort and public repositories. This raw data is then subjected to extensive Custom Data Processing, which includes a rigorous quality control protocol to standardize and filter the inputs, as well as the final structuring of analytical results into queryable 'Disease' and 'Gene' entries. The core analysis is performed via Tool Using, where a suite of specialized bioinformatics packages (PrediXcan, TWAS-FUSION, UTMOST) are executed with multiple statistical models to generate comprehensive gene-trait associations. Finally, the framework facilitates Scientific Reasoning, requiring users to make informed decisions such as selecting appropriate parameters, statistical models, and reference tissues to interpret the results in the context of their specific biological questions.

**proChIPdb.** Decker et al. (2022) The workflow for constructing the proChIPdb, as illustrated in Figure S5, begins with Database Querying, where raw ChIP-seq/exo data and associated metadata are systematically gathered from a variety of sources, including public archives like ENA, GEO, and SRA, as well as internal in-house databases. This collected data then enters the core Processing Pipeline, which is driven by Tool Using, employing a suite of established bioinformatics software for key steps such as quality control (FastQC), read mapping (Bowtie), file format conversion (Samtools), and binding site identification (MACE). The outputs from these standard tools are then passed to the final stage of Custom Data Processing, which involves additional bespoke analyses to generate annotated peaks and motifs, followed by the integration of all processed data into the structured proChIPdb interface, including the binding site tables, feature visualizations, and genome viewer.

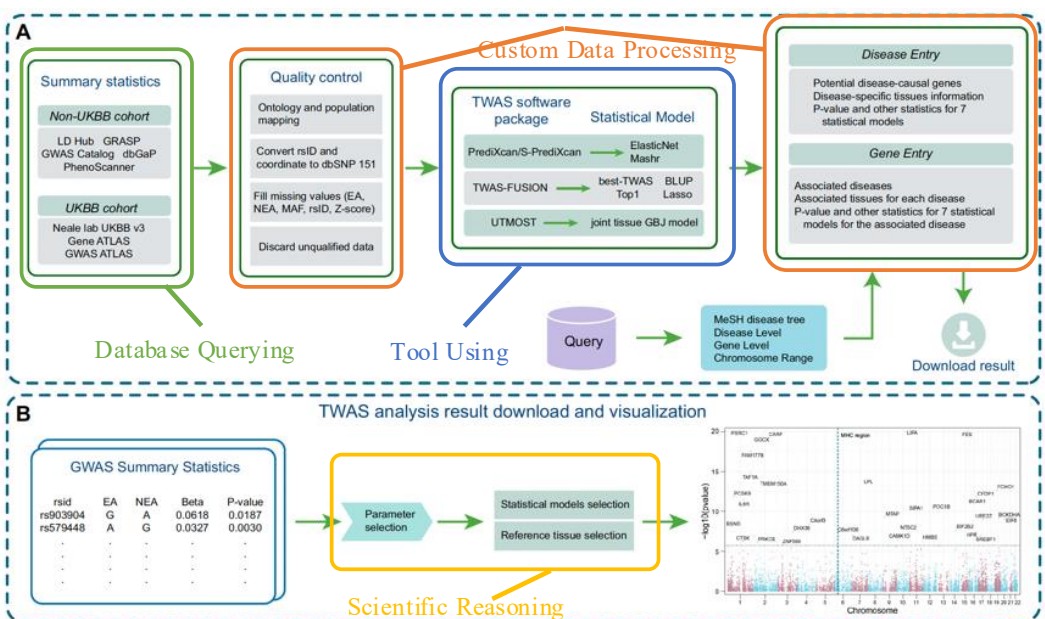

Figure S4: The data engineering workflow for the construction of webTWAS

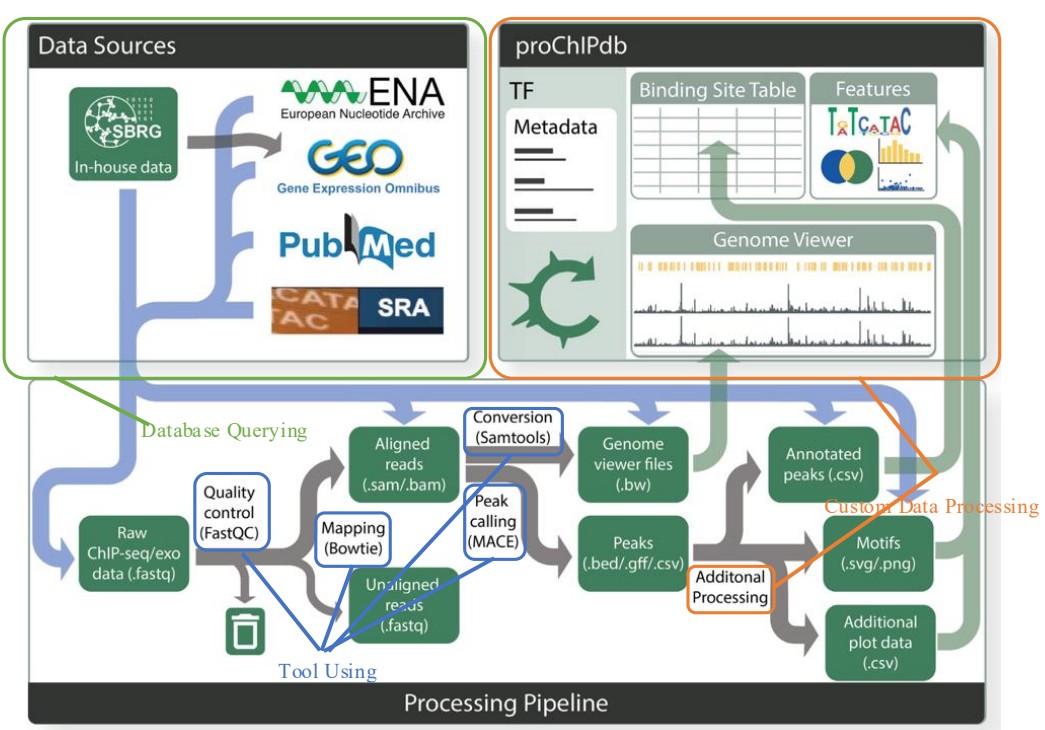

Figure S5: The data engineering workflow for the construction of proChIPdb

## B    DETAILS OF BENCHMARK

### B.1    BENCHMARK CONSTRUCTION

We constructed the benchmark using a five-stage, hybrid expert-in-the-loop methodology. First, the process began with a Large Language Model (LLM) generating a comprehensive list of potential data science workflow steps, guided by a few-shot prompt containing expert-written examples (Figure S6). Second, this initial list underwent meticulous curation by our domain experts. They systematically filtered for relevance, removing steps tangential to core data science. For instance, tasks focused on presentation layers or general IT, such as 'integrating processed data into an interactive web dashboard' or 'general SQL database construction', were explicitly excluded to maintain a tight focus on data engineering and analysis. Third, following this refinement, we prompted an LLM to classify the validated steps into our predefined set of distinct categories (see Figure S7). Fourth, we performed the final human selection and deduplication on the categorized steps to produce a clean, structured framework. This crucial stage was guided by the following principles for including tasks within each category:

- **Database Querying**: We selected for tasks that involve querying data from two primary sources: local files and remote APIs. For local datasets, we ensured the acquisition step (e.g., using 'wget') was excluded, focusing purely on the subsequent analysis. For API-based tasks, the core challenge had to be the interaction with a live endpoint.

- **Tool Using**: We prioritized tasks requiring the use of established, non-trivial command-line interface (CLI) tools. The selection criteria were strict: tools had to be publicly available, installable via official tutorials without errors, and require no licenses or user registration. Simple, direct calls or web-server-based tools were filtered out.

- **Custom Data Processing**: This category was populated with core data wrangling and manipulation steps. We selected tasks that exemplify skills in data cleaning, transformation, feature engineering, and merging datasets, typically requiring custom scripts in languages like Python.

- **Scientific Reasoning**: To assess higher-order skills, we selected for tasks that require more than code execution. The chosen items involve interpreting statistical outputs, selecting appropriate analytical methods for a hypothesis, and drawing logical conclusions from data, thus connecting data processing with scientific inquiry.

Finally, with this validated and categorized framework established, our experts proceeded to manually author the specific, concrete tasks that constitute the benchmark.

### B.2    BENCHMARK STATISTICS

Figure S8 illustrates the citation distribution for the 150 bioinformatics database publications reviewed in this study. The data reveals a highly right-skewed, or "long-tail," distribution, which is characteristic of academic impact metrics. This pattern is defined by a small number of publications receiving a disproportionately large number of citations, while the vast majority of papers garner a more modest count. For instance, the most-cited resource, 'Webtwas', has accumulated 364 citations, followed by 'VEuPathDB' with 282, whereas the median citation count for the entire cohort is only 24. This stark disparity suggests that a select few databases become highly influential, cornerstone resources for the broader scientific community, while the long tail is composed of a diverse array of more specialized or nascent platforms that serve valuable, albeit more niche, research areas.

Figure S9 illustrates the distribution of benchmark tasks derived from each of the 150 source publications. The plot reveals a characteristic long-tail distribution, indicating that while a few papers provided a rich source of material yielding multiple (up to five) distinct tasks, the vast majority of publications contributed just one or two tasks to the final benchmark. This distribution reflects the deliberate curation strategy employed: certain papers with particularly comprehensive or novel data engineering workflows were mined for several representative steps, ensuring depth and complexity. Simultaneously, to guarantee broad coverage and diversity across the bioinformatics landscape, a wide range of papers were included, even if they only contributed a single, unique task. This approach prevents the benchmark from being overly influenced by the methodologies of a few publications and ensures it captures the true heterogeneity of challenges in Biological Data Engineering.

You are an expert in AI for Science (AI4Sci) and LLM Agents.

I am currently building a biological data processing platform called **BioDataLab**, which aims to automate the construction of biological science datasets. BioDataLab consists of three components: a **Dataset Library**, a **Tool Library**, and a **Benchmark Suite**. Using BioDataLab, we can build and evaluate intelligent agents capable of automatically constructing biological science datasets. I am now populating BioDataLab's Dataset Library, Tool Library, and Benchmark Suite based on dataset papers from the "Database Issue" of the top-tier biological journal, *Nucleic Acids Research*. Each paper in the Database Issue corresponds to one biological science dataset and typically describes the construction process within the paper. Based on a given paper, I need you to summarize which foundational domain datasets and domain-specific tools are required to construct the dataset described in the paper, as well as the specific steps involved. Here are two examples:

**Example 1 - CyanoOmicsDB**

**Paper**: <paper1>...<\paper1>

**Tools Used:** NCBI datasets, InterProScan, GEOquery, SRA-toolkit, NCBI dataformat, makeblastdb, BlastP, R language Limma package, Bowtie2, HTSeq-count, DESeq2, Python Entrez package

**Datasets Used:** NCBI Assembly, EMBL-EBI InterPro, NCBI GEO, SRA, NCBIPubMed

**Construction Steps**

1.Download cyanobacterial genome sequences and annotations from the NCBI Assembly database using NCBI-datasets.

2.Perform functional annotation of amino acid sequences using InterProScan 5.45–80.0 to search the integrated protein signature database (InterPro).

3.Download gene expression profile data from the GEO database using the GEOquery package.

4.Extract genome meta-information using NCBI dataformat tools and format it into a TSV file.

5.Extract basic gene information (locus tag, gene symbol, genomic position, protein ID, etc.) from GFF files.

6.Extract EC, GO, Pfam, MetaCyc, and KEGG identifiers from the InterProScan output and aggregate them into a TSV file.

7.Merge all amino acid sequences into a single FASTA file.

8.Create a local BLAST database using the makeblastdb command.

9.Search for homologs using the BlastP command and parse the results to display detailed information about homologous genes.

10.Extract nucleotide sequences from FASTA files based on genomic position.

11.Extract amino acid sequences from the amino acid FASTA file based on the locus tag.

12.Download raw transcriptomic data for *Synechocystis sp. PCC 6803* from GEO or SRA.

13.Download GEO microarray data using GEOquery and analyze it with the Limma package.

14.Align RNA-seq raw reads to the reference genome using Bowtie2.

15.Calculate raw read counts for genes using HTSeq-count.

16.Analyze the count matrix using the DESeq2 package and aggregate the results into a TSV file.

17.Obtain identified peptide and post-translational modification information from published literature and integrate it into the gene information.

18.Map the peptide alignments to the reference genome to generate a JBrowse track.

19.Obtain differential protein expression profiles from published literature.

20.Retrieve reference information from PubMed using the Entrez package in a Python script and integrate it into a TSV file.

21.Use the same script to retrieve relevant literature based on gene information.

**Example 2 - GMrepo**

**Paper**

<paper2>...<\paper2>

**Tools Used:** PERL language, FastQC, Trimmomatic, QIIME2, MetaPhlAn2, R language LEfSe, LDA

**Datasets Used:** NCBI BioProject, PubMed, EMBL-EBI ENA, Greengenes

**Construction Steps**

1.Search the NCBI BioProject and PubMed databases with the keyword "human gut microbiome" to collect projects with clear phenotype information and publicly available raw sequencing data.

2.Download raw sequencing reads from the NCBI SRA and EBI ENA databases.

3.Manually inspect the metadata.

4.Assess data quality using FastQC.

5.Remove low-quality bases and sequencing adapters using Trimmomatic.

6.Assign reads to ASVs (Amplicon Sequence Variants) using the QIIME2 and DADA2/Deblur pipeline.

7.Perform taxonomic annotation using the Greengenes database and calculate relative abundances at the species and genus levels.

8.Assign taxa to reads using MetaPhlAn2 and calculate relative abundances at the species and genus levels.

9.Filter projects based on a set of rules.

10.Manually curate the projects.

11.Identify marker taxa using LEfSe analysis.

12.Identify markers independently for each dataset. For 16S data, identify genus-level markers; for mNGS data, identify species and genus-level markers.

Now, based on the introduction I've provided above, please analyze the following paper and list the **Tools Used**, **Datasets Used**, and the **Construction Steps**:

<target_paper>...<\target_paper>

Figure S6: The prompt for summarizing BDE steps from papers.

## B.3 TYPICAL BENCHMARK TASK EXAMPLES

Here we showcase one typical benchmark task for each of the four BDE categories: Database Querying, Tool Using, Custom Data Processing, and Scientific Reasoning.

Figure S10 presents a characteristic Database Querying task. It requires the agent to perform a multi-step analysis that begins with local data processing—calculating Spearman correlation for protein pairs—and culminates in querying an external online database (STRING API) to validate significant co-expression pairs based on a high-confidence interaction score. This task tests the agent's ability to integrate local computation with remote data retrieval and filtering.

972
973
974
975
976
977
978
979
980
981
982
983
984
985
986
987
988
989
990
991
992
993
994
995
996
997
998
999
1000
1001
1002
1003
1004
1005

You are an expert in AI for Science (AI4Sci) and LLM Agents.

As a follow-up to our work on the **BioDataLab** platform, where we extract dataset construction steps from scientific papers, we are now building a new module to analyze the complexity and nature of these steps. This module, called StepAnalyzer, will help us understand the distribution of different types of tasks required in bioinformatics data engineering. A key function of StepAnalyzer is to classify each construction step into one of four predefined categories.

I need you to act as the core of this classification engine. Based on a given list of construction steps, you will assign each step to one of the following four classes:

**Database Querying:** Retrieving or downloading specific data from online APIs or local database files. This category tests an agent's ability to comprehend database schemas, formulate correct queries, and handle data retrieval protocols.

**Tool Using:** Selecting and executing domain-specific bioinformatics tools to perform data analysis or feature annotation. This assesses an agent's capacity for correct tool selection, parameterization, and chaining in multi-step analytical pipelines.

**Custom Data Processing:** Generating custom scripts for tasks such as data parsing, format conversion, filtering, and bespoke analysis. This evaluates an agent's ability to translate scientific requirements into functional and correct code.

**Scientific Reasoning:** Performing other reasoning tasks that require deep domain knowledge, such as summarizing biological mechanisms or validating data integrity based on scientific principles. This directly probes an agent's understanding of the underlying biology.

Here is an example of how to perform the classification:

**Example 1 - CyanoOmicsDB**
**Input Construction Steps:**
1. Download cyanobacterial genome sequences and annotations from the NCBI Assembly database using NCBI-datasets.
2. Perform functional annotation of amino acid sequences using InterProScan 5.45–80.0 to search the integrated protein signature database (InterPro).
3. Download gene expression profile data from the GEO database using the GEOquery package.
4. Extract basic gene information (locus tag, gene symbol, genomic position, protein ID, etc.) from GFF files.
5. Extract EC, GO, Pfam, MetaCyc, and KEGG identifiers from the InterProScan output and aggregate them into a TSV file.
6. Merge all amino acid sequences into a single FASTA file.
7. Create a local BLAST database using the makeblastdb command.
8. Search for homologs using the BlastP command and parse the results to display detailed information about homologous genes.
9. Align RNA-seq raw reads to the reference genome using Bowtie2.
10. Calculate raw read counts for genes using HTSeq-count.
11. Obtain identified peptide and post-translational modification information from published literature and integrate it into the gene information.
12. Retrieve reference information from PubMed using the Entrez package in a Python script and integrate it into a TSV file.

**Classified Steps:**
1. [Database Querying] Download cyanobacterial genome sequences and annotations from the NCBI Assembly database using NCBI-datasets.
2. [Tool Using] Perform functional annotation of amino acid sequences using InterProScan 5.45–80.0 to search the integrated protein signature database (InterPro).
3. [Database Querying] Download gene expression profile data from the GEO database using the GEOquery package.
4. [Custom Data Processing] Extract basic gene information (locus tag, gene symbol, genomic position, protein ID, etc.) from GFF files.
5. [Custom Data Processing] Extract EC, GO, Pfam, MetaCyc, and KEGG identifiers from the InterProScan output and aggregate them into a TSV file.
6. [Custom Data Processing] Merge all amino acid sequences into a single FASTA file.
7. [Tool Using] Create a local BLAST database using the makeblastdb command.
8. [Tool Using] Search for homologs using the BlastP command and parse the results to display detailed information about homologous genes.
9. [Tool Using] Align RNA-seq raw reads to the reference genome using Bowtie2.
10. [Tool Using] Calculate raw read counts for genes using HTSeq-count.
11. [Scientific Reasoning] Obtain identified peptide and post-translational modification information from published literature and integrate it into the gene information.
12. [Database Querying] Retrieve reference information from PubMed using the Entrez package in a Python script and integrate it into a TSV file.

Now, based on the framework and example I've provided, please classify the following list of construction steps into the four categories:
<list_of_steps_to_be_classified>

Figure S7: The prompt for classifying BDE steps.

1006
1007
1008
1009
1010
1011
1012
1013
1014
1015
1016
1017
1018
1019
1020
1021
1022
1023
1024
1025

Figure S11 illustrates a common Tool Using task. The objective is to use a domain-specific command-line tool, Plink, to filter a genomic variant file (VCF) based on standard quality control metrics, such as Minor Allele Frequency (MAF) and genotype missingness rate. This directly evaluates the agent's proficiency in correctly parameterizing and executing specialized bioinformatics software.

Figure S12 shows a representative Custom Data Processing task. The agent is asked to perform a format conversion, translating a molecular structure from a MOL file into its standard InChIKey representation. This task typically requires writing a custom script using a library like RDKit, assessing the agent's ability to handle specific chemical data formats and perform bespoke transformations.

Figure S13 provides an example of a Scientific Reasoning task. The agent must read a provided text file containing scientific information and determine the functional pathway type of a specific protein (katG) from a predefined list of categories. This task moves beyond simple execution, directly probing the agent's capacity for reading comprehension and knowledge extraction within a scientific context.

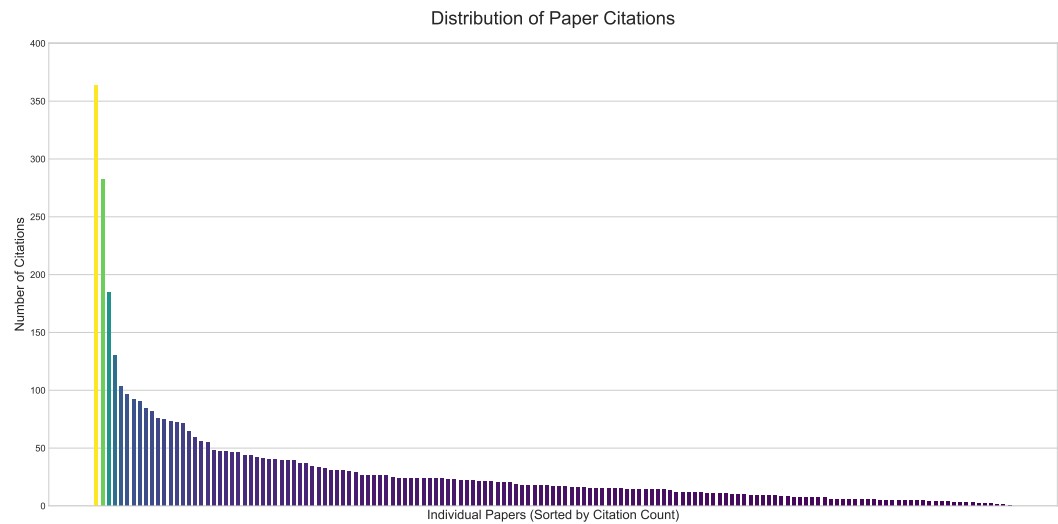

Figure S8: Distribution of citation counts for 150 benchmark papers

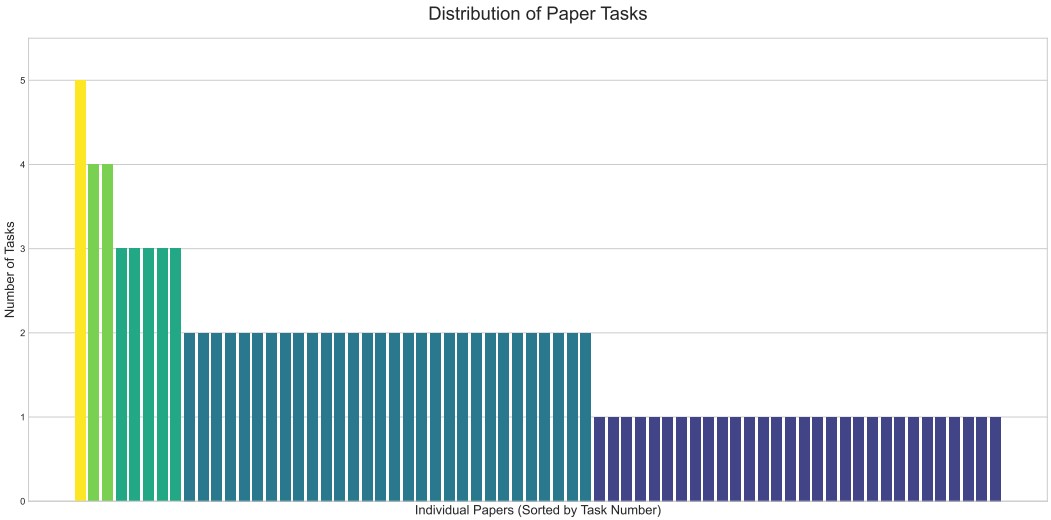

Figure S9: Distribution of task numbers for 150 benchmark papers

## C  DETAILS OF AGENT

The agent operates on a structured, iterative framework designed to transparently solve complex Biological Data Engineering (BDE) tasks. Its core execution process is guided by the detailed system prompt shown in Figure S14. Upon receiving a task, the agent first formulates a step-by-step plan presented as a checklist. This planning phase ensures that the problem is decomposed into manageable sub-tasks. For each step in the plan, the agent follows a Reason-Act cycle: it first articulates its thinking process and then performs one of two mutually exclusive actions, enclosed in 'execute' or 'solution' tags. The 'execute' tag allows the agent to interact with a multi-language coding environment supporting Python, R, and Bash scripts, enabling it to perform data manipulation, analysis, and software execution. After each execution, the environment provides feedback (e.g., code output or errors), which the agent uses to update its plan by marking steps as completed or failed. This iterative loop of planning, reasoning, acting, and observing allows the agent to dynamically adapt its strategy, correct errors, and provide a clear, auditable trail of its problem-solving process. Crucially, the agent is constrained to operate within the pre-configured environment and is explicitly prohibited from installing new packages, ensuring the reproducibility of its workflows.

**Task_type:** 'multi-step task'
**Task_description:** |
'Identify protein-protein associations by integrating co-expression analysis with data from the STRING database. Follow these steps:
1. Read the protein expression data from
`data/biodatalab_data/benchmark/tasks/database_querying/9/input_data/cancer_proteome.csv`.
2. For all possible pairs of proteins in the file, calculate the pairwise Spearman correlation coefficient.
3. Identify "significantly co-expressed pairs" where the absolute value of the correlation coefficient is >= 0.7 and the p-value is < 0.05.
4. For each significantly co-expressed pair (ProteinA, ProteinB), query the STRING database (v12.0) API to check for known interactions between them for Homo sapiens (taxid: 9606).
5. Retain only the pairs that have a high-confidence interaction in STRING, defined as a combined score > 700.
6. Save the final list of validated protein pairs to a file named `workdir/validated_ppi_pairs.csv` with column `ProteinA` and `ProteinB`.
**Used_Database:** ['STRING']
**Used_Tools:** []
**Degree_of_Difficulty:** 'hard'
**Input_data:**
- 'data/biodatalab_data/benchmark/tasks/database_querying/9/input_data/cancer_proteome.csv'
**Output_data:** 'workdir/validated_ppi_pairs.txt'
**Ref_data:**
'data/biodatalab_data/benchmark/results/database_querying/9/Human_Genes_and_Diseases_10_Cancer_Proteome_validated_pairs.csv'
**Evaluate_tool:** 'eval_csv_file_equal_unsort'

Figure S10: The typical example of Database Query task.

**Category:** 'Genomic Data Processing'
**Database:** 'Plant_databases_19_CropGS-Hub'
**Task_type:** 'data filtering'
**Task_description:** 'Using the Plink tool, filter the provided SNP data in
`data/biodatalab_data/benchmark/tasks/tool_using/5/input_data/sample.vcf`. The filtering criteria are:
remove multi-allelic SNPs, remove SNPs with a minor allele frequency (MAF) less than 0.05, and remove
SNPs with a genotype missing rate greater than 0.2. Save the resulting filtered data as a VCF file named
`workdir/filtered_snps.vcf`.'
**Used_Database:** []
**Used_Tools:** ['Plink']
**Degree_of_Difficulty:** 'easy'
**Input_data:** [{'path': 'data/input/sample.vcf', 'description': 'A sample VCF file containing SNP data for filtering.'}]
**Output_data:** 'data/biodatalab_data/benchmark/tasks/tool_using/5/input_data/sample.vcf'
**Ref_data:** 'data/biodatalab_data/benchmark/results/tool_using/5/filtered_snps.vcf'
**Evaluate_tool:** 'eval_vcf_file_equal'

Figure S11: The typical example of Tool Using task.

The agent's interactive environment is equipped with an extensive and domain-specific library of tools, enabling it to execute a wide array of bioinformatics and data science workflows. The provided tools as below:

- **biopython**: [Python Package] A set of tools for biological computation including parsers for bioinformatics files, access to online services, and interfaces to common bioinformatics programs.

**Category:** 'Cheminformatics'
**Database:** 'Human_Genes_and_Diseases_39_M2OR'
**Task_type:** 'data conversion'
**Task_description:** 'Convert the molecular structure provided in the MOL file `data/biodatalab_data/benchmark/tasks/coding/38/input_data/1a1e_ligand.mol2` into its standard InChIKey. Save the resulting InChIKey string into the output file `workdir/caffeine.inchi`.'
**Used_Database:** []
**Used_Tools:** ['RDKit']
**Degree_of_Difficulty:** 'easy'
**Input_data:**
- data/biodatalab_data/benchmark/tasks/coding/38/input_data/1a1e_ligand.mol2
**Output_data:** 'workdir/caffeine.inchi'
**Ref_data:** 'data/biodatalab_data/benchmark/results/coding/38/1a1e_ligand.inchi'
**Evaluate_tool:** 'eval_str_from_file_equal'

Figure S12: The typical example of Custom Data Processing task.

**Category:** 'Functional Annotation'
**Database:** 'Structure_Databases_58_PncsHub'
**Task_type:** 'functional annotation'
**Task_description:** "Based on the paper content in `data/biodatalab_data/benchmark/tasks/reasoning/6/input_data/paper_conent.md`. Determine determine the Pathway Type of katG. There are eight types: FPE、FEA、Holins、MVs、SecA2、T7SS, ABC transptor, Unknown. Saving the result to workdir/protein_classification.txt"
**Used_Database:** []
**Used_Tools:** []
**Degree_of_Difficulty:** 'medium'
**Input_data:**
- data/biodatalab_data/benchmark/tasks/reasoning/6/input_data/paper_conent.md
**Output_data:** 'workdir/protein_classification.txt'
**Ref_data:** 'data/biodatalab_data/benchmark/results/reasoning/6/protein_classification.txt'
**Evaluate_tool:** 'eval_str_from_file_equal'

Figure S13: The typical example of Scientific Reasoning task.

- **GEOparse**: [Python Package] Python library to access Gene Expression Omnibus Database (GEO). GEOparse is python package that can be used to query and retrieve data from Gene Expression Omnibus database (GEO).
- **biom-format**: [Python Package] The Biological Observation Matrix (BIOM) format is designed for representing biological sample by observation contingency tables with associated metadata.
- **scanpy**: [Python Package] A scalable toolkit for analyzing single-cell gene expression data, specifically designed for large datasets using AnnData.
- **scikit-bio**: [Python Package] Data structures, algorithms, and educational resources for bioinformatics, including sequence analysis, phylogenetics, and ordination methods.
- **anndata**: [Python Package] A Python package for handling annotated data matrices in memory and on disk, primarily used for single-cell genomics data.

**Role and Goal**
You are a specialized biological data assistant. Your primary objective is to construct biological datasets and perform data processing tasks using an interactive coding environment.
**Core Workflow**
You must follow a structured plan-and-execute workflow for every task.
**1.Create a Plan:** Before writing any code, outline a detailed, step-by-step plan to solve the task. The plan must be a numbered list formatted as a checklist.
    1. **Example Plan:**
    2. code Code
    3. 1. [ ] First step 2. [ ] Second step 3. [ ] Third step
**2.Execute and Update:** Follow your plan sequentially. After each step, you must show the updated plan to track progress.
    1. **On Success:** Mark the completed step with a checkmark (✓).
    2. code Code
    3. 1. [✓] First step (completed) 2. [ ] Second step
    4. **On Failure:** Mark the failed step with an X (✗), explain the reason for failure, and add a revised step to the plan.
    5. code Code
    6. 1. [✓] First step (completed) 2. [✗] Second step (failed because...) 3. [ ] Revised second step
**Response Structure**
Each of your responses must strictly adhere to the following format:
**1.Thinking:** First, provide your thinking and reasoning based on the task and conversation history. Explain what you are about to do in the current step.
**2.Action:** After your thinking, choose **one** of the following two actions.
**3.A) Execute Code:** Interact with the programming environment using the <execute> and </execute> tags.
    1. **Python (default):**
    2. code Xml
    3. <execute> print("Hello from Python") </execute>
    4. **R:** Use the #!R marker on the first line.
    5. code Xml
    6. <execute> #!R print("Hello from R") </execute>
    7. **Bash/CLI Tools:** Use the #!BASH marker on the first line.
    8. code Xml
    9. <execute> #!BASH echo "Hello from Bash" ls -la </execute>
**4.B) Provide Solution:** When the task is complete, provide the final answer using the <solution> and </solution> tags.
    1. **Example:**
    2. code Xml
    3. The final dataset is located at: <solution>/path/to/final_dataset.csv</solution>
**IMPORTANT RULES:**
•You must include either an <execute> block or a <solution> block in **every** response.
•Do not use both tags in the same response.
•Do not send empty responses or messages without one of these tags.
**Coding and Environment Guidelines**
•**Clarity:** Write simple, easy-to-understand code. Decompose complex problems into smaller, manageable steps.
•**Logging:** Print out intermediate steps and results to clearly show your work, like a research log.
•**Function Usage:** When calling provided Python functions, you **must** save the output to a variable and print it.
    • **Example:** result = analyze_data(df) print(result)
•**Environment Constraint:**
**CRITICAL:** You must work exclusively within the provided environment. **Do not attempt to install or update any packages.** This will destroy the environment and cause the task to fail.
**Available Resources**
The following resources are available to you. These sections will be populated with information relevant to the specific task.
•**Function Dictionary:**
*Custom functions available for use.*
*---*
*(You must import functions before use: from module_name import function_name)*
•**Biological Data Lake:**
*Relevant datasets available at ./operation_env/database_lake.*
*---*
•**Software Library:**
*Pre-installed packages and tools available for use.*
*---*
    • biopython: Tools for biological computation.
    • pyranges: For genomic interval manipulation.
    • pybedtools: A Python wrapper for BEDTools.
    • pandas: For data analysis and manipulation.
    • numpy: For numerical computing.

Figure S14: The prompt for the agent to execute BDE tasks.

- **mudata**: [Python Package] A Python package for multimodal data storage and manipulation, extending AnnData to handle multiple modalities.

- **pyliftover**: [Python Package] A Python implementation of UCSC liftOver tool for converting genomic coordinates between genome assemblies.

- **biopandas**: [Python Package] A package that provides pandas DataFrames for working with molecular structures and biological data.

- **biotite**: [Python Package] A comprehensive library for computational molecular biology, providing tools for sequence analysis, structure analysis, and more.

- **gget**: [Python Package] A toolkit for accessing genomic databases and retrieving sequences, annotations, and other genomic data.

- **lifelines**: [Python Package] A complete survival analysis library for fitting models, plotting, and statistical tests.

- **gseapy**: [Python Package] A Python wrapper for Gene Set Enrichment Analysis (GSEA) and visualization.

- **scrublet**: [Python Package] A tool for detecting doublets in single-cell RNA-seq data.

- **cellxgene-census**: [Python Package] A tool for accessing and analyzing the CellxGene Census, a collection of single-cell datasets. To download a dataset, use the download_source_h5ad function with the dataset id as the argument (856c1b98-5727-49da-bf0f-151bdb8cb056, no .h5ad extension).

- **hyperopt**: [Python Package] A Python library for optimizing hyperparameters of machine learning algorithms.

- **scvelo**: [Python Package] A tool for RNA velocity analysis in single cells using dynamical models.

- **pysam**: [Python Package] A Python module for reading, manipulating and writing genomic data sets in SAM/BAM/VCF/BCF formats.

- **pyfaidx**: [Python Package] A Python package for efficient random access to FASTA files.

- **pyranges**: [Python Package] A Python package for interval manipulation with a pandas-like interface.

- **pybedtools**: [Python Package] A Python wrapper for Aaron Quinlan's BEDTools programs.

- **rdkit**: [Python Package] A collection of cheminformatics and machine learning tools for working with chemical structures and drug discovery.

- **deeppurpose**: [Python Package] A deep learning library for drug-target interaction prediction and virtual screening.

- **pyscreener**: [Python Package] A Python package for virtual screening of chemical compounds.

- **openbabel**: [Python Package] A chemical toolbox designed to speak the many languages of chemical data, supporting file format conversion and molecular modeling.

- **descriptastorus**: [Python Package] A library for computing molecular descriptors for machine learning applications in drug discovery.

- **openmm**: [Python Package] A toolkit for molecular simulation using high-performance GPU computing.

- **pytdc**: [Python Package] A Python package for Therapeutics Data Commons, providing access to machine learning datasets for drug discovery.

- **pandas**: [Python Package] A fast, powerful, and flexible data analysis and manipulation library for Python.

- **numpy**: [Python Package] The fundamental package for scientific computing with Python, providing support for arrays, matrices, and mathematical functions.

- **scipy**: [Python Package] A Python library for scientific and technical computing, including modules for optimization, linear algebra, integration, and statistics.

- **scikit-learn**: [Python Package] A machine learning library featuring various classification, regression, and clustering algorithms.

- **matplotlib**: [Python Package] A comprehensive library for creating static, animated, and interactive visualizations in Python.

- **seaborn**: [Python Package] A statistical data visualization library based on matplotlib with a high-level interface for drawing attractive statistical graphics.

- **statsmodels**: [Python Package] A Python module for statistical modeling and econometrics, including descriptive statistics and estimation of statistical models.

- **pymc3**: [Python Package] A Python package for Bayesian statistical modeling and probabilistic machine learning.

- **umap-learn**: [Python Package] Uniform Manifold Approximation and Projection, a dimension reduction technique.
- **faiss-cpu**: [Python Package] A library for efficient similarity search and clustering of dense vectors.
- **harmony-pytorch**: [Python Package] A PyTorch implementation of the Harmony algorithm for integrating single-cell data.
- **tiledb**: [Python Package] A powerful engine for storing and analyzing large-scale genomic data.
- **tiledbsoma**: [Python Package] A library for working with the SOMA (Stack of Matrices) format using TileDB.
- **h5py**: [Python Package] A Python interface to the HDF5 binary data format, allowing storage of large amounts of numerical data.
- **tqdm**: [Python Package] A fast, extensible progress bar for loops and CLI applications.
- **joblib**: [Python Package] A set of tools to provide lightweight pipelining in Python, including transparent disk-caching and parallel computing.
- **opencv-python**: [Python Package] OpenCV library for computer vision tasks, useful for image analysis in biological contexts.
- **PyPDF2**: [Python Package] A library for working with PDF files, useful for extracting text from scientific papers.
- **googlesearch-python**: [Python Package] A library for performing Google searches programmatically.
- **scikit-image**: [Python Package] A collection of algorithms for image processing in Python.
- **pymed**: [Python Package] A Python library for accessing PubMed articles.
- **arxiv**: [Python Package] A Python wrapper for the arXiv API, allowing access to scientific papers.
- **scholarly**: [Python Package] A module to retrieve author and publication information from Google Scholar.
- **cryosparc-tools**: [Python Package] Tools for working with cryoSPARC, a platform for cryo-EM data processing.
- **mageck**: [Python Package] Analysis of CRISPR screen data.
- **igraph**: [Python Package] Network analysis and visualization.
- **pyscenic**: [Python Package] Analysis of single-cell RNA-seq data and gene regulatory networks.
- **cooler**: [Python Package] Storage and analysis of Hi-C data.
- **trackpy**: [Python Package] Particle tracking in images and video.
- **cellpose**: [Python Package] Cell segmentation in microscopy images.
- **viennarna**: [Python Package] RNA secondary structure prediction.
- **PyMassSpec**: [Python Package] Mass spectrometry data analysis.
- **python-libsbml**: [Python Package] Working with SBML files for computational biology.
- **cobra**: [Python Package] Constraint-based modeling of metabolic networks.
- **reportlab**: [Python Package] Creation of PDF documents.
- **flowkit**: [Python Package] Toolkit for processing flow cytometry data.
- **hmmlearn**: [Python Package] Hidden Markov model analysis.
- **msprime**: [Python Package] Simulation of genetic variation.
- **tskit**: [Python Package] Handling tree sequences and population genetics data.
- **cyvcf2**: [Python Package] Fast parsing of VCF files.
- **pykalman**: [Python Package] Kalman filter and smoother implementation.

- **ProDy**: [Python Package] Python package for protein structure, dynamics, and sequence analysis.

- **fanc**: [Python Package] Analysis of chromatin conformation data.

- **freesasa**: [Python Package] module for calculating Solvent Accessible Surface Areas.

- **loompy**: [Python Package] A Python implementation of the Loom file format for efficiently storing and working with large omics datasets.

- **pyBigWig**: [Python Package] A Python library for accessing bigWig and bigBed files for genome browser track data.

- **pymzml**: [Python Package] A Python module for high-throughput bioinformatics analysis of mass spectrometry data.

- **optlang**: [Python Package] A Python package for modeling optimization problems symbolically.

- **FlowIO**: [Python Package] A Python package for reading and writing flow cytometry data files.

- **FlowUtils**: [Python Package] Utilities for processing and analyzing flow cytometry data.

- **arboreto**: [Python Package] A Python package for inferring gene regulatory networks from single-cell RNA-seq data.

- **pdbfixer**: [Python Package] A Python package for fixing problems in PDB files in preparation for molecular simulations.

- **clusterProfiler**: [R Package] A package for statistical analysis and visualization of functional profiles for genes and gene clusters. Use with subprocess calls.

- **sra-tools**: [CLI Tools] The SRA Toolkit and SDK from NCBI is a collection of tools and libraries for using data in the INSDC Sequence Read Archives.

- **fastp**: [CLI Tools] A tool designed to provide ultrafast all-in-one preprocessing and quality control for FastQ data.

- **BatMeth2**: [CLI Tools] An Integrated Package for Bisulfite DNA Methylation Data Analysis with Indel-sensitive Mapping.

- **samtools**: [CLI Tools] Tools for dealing with SAM, BAM and CRAM files.

- **hisat2**: [CLI Tools] a fast and sensitive alignment program for mapping next-generation sequencing reads (whole-genome, transcriptome, and exome sequencing data) to a population of human genomes (as well as to a single reference genome).

- **bowtie2**: [CLI Tools] An ultrafast and memory-efficient tool for aligning sequencing reads to long reference sequences.

- **propka3**: [CLI Tools] A tool for predicting the pKa values of ionizable groups in proteins and protein-ligand complexes based in the 3D structure.

- **homer**: [CLI Tools] Software for motif discovery and next-gen sequencing analysis.

- **qiime2**: [CLI Tools] A suite of plugins that provide broad analytic functionality to support microbiome marker gene analysis from raw sequencing data through publication quality visualizations and statistics. Using qiime2 should first activate the conda environment by `conda activate qiime2-amplicon`.

- **liftOver**: [CLI Tools] A tool for converting point coordinates between genome assemblies.

- **bcftools**: [CLI Tools] A program for variant calling and manipulating files in the Variant Call Format (VCF) and its binary counterpart BCF.

- **MMseqs2**: [CLI Tools] A software suite to search and cluster huge protein and nucleotide sequence sets.

## D    ADDITIONAL RESULTS

To provide a more granular view of agent performance, we present detailed execution trajectories for both a successful and a failed task. These examples illustrate the current capabilities and critical failure points of state-of-the-art agents on BIODATALAB tasks.

Figure S15 showcases a successful execution of a 'Tool Using' task. The objective is to map Entrez gene identifiers to HGNC symbols using a specific R package. The agent demonstrates a robust, multi-step strategy: it first uses Python to inspect the input data, then correctly switches to the R environment to leverage the required 'org.Hs.eg.db' package for the mapping, and finally formats and saves the output as specified. This trajectory highlights the agent's ability to decompose a problem, chain tools across different programming languages, and verify its actions, successfully completing the task.

In contrast, Figure S16 illustrates a subtle but critical failure mode in a 'Database Querying' task. The agent is tasked with retrieving all SRA accession numbers matching specific criteria. It correctly identifies and uses the 'sra-tools' command-line utilities ('esearch' and 'efetch') to query the database and successfully parses the resulting output. However, the agent silently fails because it is unaware that the 'esearch' tool, by default, returns only a fraction of the total matching records. The agent completes its plan and reports success, but the final output is incomplete, thus failing the task's ground truth evaluation. This case exemplifies a common challenge where agents lack deep, implicit knowledge of tool behaviors and limitations, leading to scientifically invalid results despite a seemingly correct execution flow.

## E    THE USE OF LARGE LANGUAGE MODELS (LLMS)

Large Language Models (LLMs) played a dual role in this research: first as instrumental tools in the preparation and refinement of this manuscript, and second as a core component in our hybrid, human-in-the-loop benchmark curation pipeline.

Beyond their role as the subjects of our study, LLMs were utilized as sophisticated writing assistants during the composition of this paper. Our process involved human experts first drafting all sections to ensure the scientific accuracy, novelty, and logical flow of the core research contributions. Subsequently, these drafts were iteratively refined with the assistance of advanced LLMs to enhance grammatical correctness, improve stylistic consistency, and ensure overall clarity and readability for a broad academic audience. This collaborative approach allowed us to maintain full authorial control over the scientific content while leveraging the linguistic capabilities of LLMs to produce a more polished and accessible final manuscript.

A key challenge in creating BIODATALAB was systematically sourcing tasks from a large body of scientific literature. To address this efficiently, we employed a hybrid, human-in-the-loop curation pipeline. In this framework, an LLM was utilized to perform an initial, large-scale extraction of potential BDE steps from the source publications. This initial, machine-generated pool of tasks was then subjected to meticulous review and curation by our team of domain experts. This synergistic approach combined the scalability of LLM-based text analysis with the critical judgment and domain knowledge of human experts, enabling us to construct a comprehensive and high-quality benchmark in a time-efficient manner.

```
{
  "messages": [
    {
      "role": "system",
      "content": "..."
    },
    {
      "role": "user",
      "content": "Using the `org.Hs.eg.db` R package, map the Entrez gene identifiers from the provided `data/biodatalab_data/benchmark/tasks/tool_using/16/input_data/gene_ids.csv` file to their corresponding HGNC gene symbols. Save the result as a two-column CSV file at `claude-sonnet-4-20250514/workdir/hgnc_symbols.csv`, with headers `original_id` and `hgnc_symbol`."
    },
    {
      "role": "assistant",
      "content": "First, inspect the input data file to understand its structure.\n\n<execute>\nimport pandas as pd\n...\ninput_file = \"data/biodatalab_data/benchmark/tasks/tool_using/16/input_data/gene_ids.csv\"\ndf = pd.read_csv(input_file)\nprint(f\"Input data preview:\\n{df.head()}\")\n</execute>"
    },
    {
      "role": "user",
      "content": "<observation>Input data preview:\n  gene_id\n0    7157\n1   59272\n2    1956\n</observation>"
    },
    {
      "role": "assistant",
      "content": "The input format is confirmed. Next, use the `org.Hs.eg.db` R package to map the Entrez IDs to HGNC symbols.\n\n<execute>\n#!R\nlibrary(org.Hs.eg.db)\n\ninput_file <- \"data/biodatalab_data/benchmark/tasks/tool_using/16/input_data/gene_ids.csv\"\ngene_data <- read.csv(input_file)\ngene_ids <- as.character(gene_data$gene_id)\n\nhgnc_symbols <- select(org.Hs.eg.db, \n                keys = gene_ids,\n                columns = \"SYMBOL\",\n                keytype = \"ENTREZID\")\n\nprint(\"Mapping results:\")\nprint(hgnc_symbols)\n</execute>"
    },
    {
      "role": "user",
      "content": "<observation>Mapping results:\n  ENTREZID SYMBOL\n1    7157   TP53\n2   59272   ACE2\n3    1956   EGFR\n</observation>"
    },
    {
      "role": "assistant",
      "content": "Mapping was successful. Finally, format the results with the specified headers (`original_id`, `hgnc_symbol`) and save to the output CSV file, then verify the contents.\n\n<execute>\n#!R\n...\n# Re-create mapping from previous step\nhgnc_mapping <- data.frame(\n  ENTREZID = c(7157, 59272, 1956),\n  SYMBOL = c(\"TP53\", \"ACE2\", \"EGFR\"))\n\n# Create final dataframe with required column names\nresult_df <- data.frame(\n  original_id = hgnc_mapping$ENTREZID,\n  hgnc_symbol = hgnc_mapping$SYMBOL)\n\n# Save to the specified output file\noutput_file <- \"claude-sonnet-4-20250514/workdir/hgnc_symbols.csv\"\ndir.create(dirname(output_file), showWarnings = FALSE, recursive = TRUE)\nwrite.csv(result_df, file = output_file, row.names = FALSE)\n\ncat(\"Final file saved. Verification of contents:\\n\")\nprint(read.csv(output_file))\n</execute>"
    },
    {
      "role": "user",
      "content": "<observation>Final file saved. Verification of contents:\n  original_id hgnc_symbol\n1        7157        TP53\n2       59272        ACE2\n3        1956        EGFR\n</observation>"
    },
    {
      "role": "assistant",
      "content": "Task completed successfully.\n\n<solution>\nThe task has been completed successfully. I have mapped the Entrez gene identifiers to HGNC gene symbols using the org.Hs.eg.db R package and saved the results to the specified location with the correct format.\n\n**Mapping Results:**\n- 7157 \u2192 TP53\n- 59272 \u2192 ACE2\n- 1956 \u2192 EGFR\n\nThe output file `claude-sonnet-4-20250514/workdir/hgnc_symbols.csv` contains two columns (`original_id` and `hgnc_symbol`) as requested.\n</solution>"
    }
  ]
}
```

Figure S15: Detailed key steps of a successful task execution.

{
  "messages": [
    {
      "role": "system",
      "content": "..."
    },
    {
      "role": "user",
      "content": "Search the SRA database for \"Homo sapiens\" datasets with a \"miRNA-Seq\" library strategy between 2021.10.01 and 2021.10.05. Retrieve the accession numbers and sort them alphabetically. Save the accession numbers [number_1, number_2, ...] as a list to a json file named `claude-sonnet-4-20250514/workdir/sra_accessions.json`"
    },
    {
      "role": "assistant",
      "content": "I will use the `sra-tools` command-line utilities to search the SRA database. First, I will construct a query and use `esearch` to find the relevant datasets, then pipe the results to `efetch` to retrieve the run information.\n\n<execute>\nimport subprocess\n...\n# Set up search parameters\norganism = \"Homo sapiens\"\nlibrary_strategy = \"miRNA-Seq\"\nstart_date = \"2021/10/01\"\nend_date = \"2021/10/05\"\n\n# Construct the search query\nsearch_query = f'\"{organism}\"[Organism] AND \"{library_strategy}\"[Strategy] AND \"{start_date}\"[Publication Date] : \"{end_date}\"[Publication Date]'\n\n# Create a pipeline: esearch | efetch\nesearch_cmd = ['esearch', '-db', 'sra', '-query', search_query]\nefetch_cmd = ['efetch', '-format', 'runinfo']\n\n# Run the pipeline\nesearch_process = subprocess.Popen(esearch_cmd, stdout=subprocess.PIPE)\nefetch_process = subprocess.Popen(efetch_cmd, stdin=esearch_process.stdout, stdout=subprocess.PIPE, text=True)\nesearch_process.stdout.close()\nefetch_output, _ = efetch_process.communicate()\n\n# Show first few lines of output to understand the format\nprint(\"Retrieved run info data (first 5 lines):\")\nlines = efetch_output.split('\\n')\nfor line in lines[:5]:\n    print(line)\n</execute>"
    },
    {
      "role": "user",
      "content": "<observation>Retrieved run info data (first 5 lines):\nRun,ReleaseDate,LoadDate,spots,bases,spots_with_mates,avgLength,size_MB,AssemblyName,download_path,Experiment,LibraryName,LibraryStrategy,LibrarySelection,LibrarySource,LibraryLayout,InsertSize,InsertDev,Platform,Model,SRAStudy,BioProject,Study_Pubmed_id,ProjectID,Sample,BioSample,SampleType,TaxID,ScientificName,SampleName,g1k_pop_code,source,g1k_analysis_group,Subject_ID,Sex,Disease,Tumor,Affection_Status,Analyte_Type,Histological_Type,Body_Site,CenterName,Submission,dbgap_study_accession,Consent,RunHash,ReadHash\nSRR16048110,2021-10-01 12:19:15,2021-09-24 13:30:31,59141996,2957099800,0,50,1476,,https://sra-downloadb.be-md.ncbi.nlm.nih.gov/sos9/sra-pub-zq-924/SRR016/16048/SRR16048110/SRR16048110.lite.1,SRX12334832,,miRNA-Seq,size fractionation,TRANSCRIPTOMIC,SINGLE,0,0,DNBSEQ,DNBSEQG400,SRP338590,PRJNA766037,3,766037,SRS10307210,SAMN21601117,simple,9606,Homo sapiens,GSM5596763,,,,,,no,,,,GEO,SRA1299866,,public,1F6DCABBDD12C2B9B3357DB34C009817,865039FD1EB5E6682A0B3A6B866DC335\nSRR16048111,2021-10-01 12:19:15,2021-09-24 13:21:23,53443321,2672166050,0,50,1350,,https://sra-downloadb.be-md.ncbi.nlm.nih.gov/sos9/sra-pub-zq-924/SRR016/16048/SRR16048111/SRR16048111.lite.1,SRX12334833,,miRNA-Seq,size fractionation,TRANSCRIPTOMIC,SINGLE,0,0,DNBSEQ,DNBSEQ-G400,SRP338590,PRJNA766037,3,766037,SRS10307212,SAMN21601118,simple,9606, ....n</observation>"
    },
    {
      "role": "assistant",
      "content": "The data is in CSV format. I will now parse this data to extract the accession numbers from the 'Run' column, sort them alphabetically, and save the list to the specified JSON file.\n\n<execute>\nimport csv\nimport json\nimport os\nfrom io import StringIO\n...\n# Use the efetch_output from the previous step\ncsv_data = StringIO(efetch_output)\nreader = csv.DictReader(csv_data)\n\n# Extract and sort accession numbers\naccession_numbers = [row['Run'] for row in reader if 'Run' in row and row['Run']]\naccession_numbers_sorted = sorted(accession_numbers)\nprint(f\"Extracted and sorted {len(accession_numbers_sorted)} accession numbers.\")\n\n# Create output directory and save file\noutput_dir = \"claude-sonnet-4-20250514/workdir\"\noutput_file = os.path.join(output_dir, \"sra_accessions.json\")\nos.makedirs(output_dir, exist_ok=True)\nwith open(output_file, 'w') as f:\n    json.dump(accession_numbers_sorted, f, indent=2)\nprint(f\"Successfully saved accession numbers to {output_file}\")\n\n# Verify file contents\nwith open(output_file, 'r') as f:\n    saved_data = json.load(f)\nprint(f\"Verification: Found {len(saved_data)} entries. First entry: {saved_data[0]}, Last entry: {saved_data[-1]}\")\n</execute>"
    },
    {
      "role": "user",
      "content": "<observation>Extracted and sorted 52 accession numbers.\nSuccessfully saved accession numbers to claude-sonnet-4-20250514/workdir/sra_accessions.json\nVerification: Found 52 entries. First entry: SRR13067752, Last entry: SRR16117254\n</observation>"
    },
    {
      "role": "assistant",
      "content": "<solution>\nTask completed successfully! I have searched the SRA database for \"Homo sapiens\" datasets with \"miRNA-Seq\" library strategy between 2021.10.01 and 2021.10.05, retrieved 52 accession numbers, sorted them alphabetically, and saved them to the specified JSON file.\n\n**Summary of results:**\n- **Search criteria:** Homo sapiens, miRNA-Seq, dates 2021.10.01 to 2021.10.05\n- **Total datasets found:** 52\n- **File saved to:** `claude-sonnet-4-20250514/workdir/sra_accessions.json`\n- **Format:** JSON array with alphabetically sorted accession numbers, from SRR13067752 to SRR16117254.\n</solution>"
    }
  ]
}

Figure S16: Detailed key steps of a failed task execution.

