# OpenReview forum: "BioDataLab: Towards Generalist Agents for Real-world Biological Data Engineering"
_ICLR.cc/2026/Conference — Submitted to ICLR 2026_

### Official Review · Reviewer_Beei · 2025-10-28

**Soundness:** 3
**Presentation:** 3
**Contribution:** 3
**Rating:** 6
**Confidence:** 4

**Summary:**

The paper introduces BIODATALAB, the first benchmark targeting Biological Data Engineering (BDE) — the process of constructing and curating biological datasets that underpin genomics, bioinformatics, and drug discovery. The benchmark is composed of 114 real-world tasks curated from 150 Nucleic Acids Research (NAR) papers, covering four categories: database querying, tool using, custom data processing, and scientific reasoning. BIODATALAB emphasizes three core design principles: (1) managing procedural ambiguity, (2) establishing intermediate ground truth through tractable data replication, and (3) enabling multi-modal evaluation using custom, domain-aware evaluators that go beyond text matching.

Experiments benchmark state-of-the-art LLM agents (GPT-4.1, Claude 4, Gemini 2.5, DeepSeek, Qwen 3) within a common agent framework (Biomni). Results show that even the strongest models achieve modest success rates (GPT-4.1 tops at 38.5%), highlighting the difficulty of automating complex scientific workflows. The authors identify several key failure modes — brittle tool chaining, hallucinated tool parameters, poor handling of specialized file formats, and weak long-horizon reasoning. The paper positions BIODATALAB as both a rigorous evaluation suite and a roadmap for developing more capable “scientific agents.”

**Strengths:**

1. The paper formalizes Biological Data Engineering (BDE) as a standalone AI challenge, bridging a gap between data-science automation and domain-specific bioinformatics. This conceptual framing is original and well-motivated.

2. Tasks are directly derived from real scientific workflows (NAR datasets), ensuring realism and relevance to actual biological research. Additionally, the authors clearly outline a multi-stage curation pipeline combining human expertise and systematic criteria for diversity, representativeness, and tractability. Their inclusion of manually replicated ground-truth outputs lends credibility and reproducibility.

3. Empirical results and analysis identify interpretable failure patterns, offering diagnostic insights into current agent weaknesses.

**Weaknesses:**

1. How were the evaluation functions curated? Are they the same people who collect the tasks and curate the evaluation functions? If so, there might be potential bias in the evaluation functions.

2. The paper doesn't compare the agents' performance with traditional methods or domain-specific tools, making it difficult to assess their practical utility relative to existing solutions. Including such comparisons would provide valuable context to evaluate the agents' real-world usefulness and guide future improvements.

3. There lack of a human study to show the relevance of the eval functions to the real task success.

**Questions:**

1. There is another related data-driven scientific coding benchmark might be missed in the related benchmarks: "ScienceAgentBench: Toward Rigorous Assessment of Language Agents for Data-Driven Scientific Discovery"

---

### Official Review · Reviewer_iMuJ · 2025-10-31

**Soundness:** 2
**Presentation:** 2
**Contribution:** 1
**Rating:** 2
**Confidence:** 4

**Summary:**

This paper presents BioDataLab, a benchmark of 114 biological data engineering tasks sourced from research publications. Experimental results show that LLMs struggle at tasks that require deep scientific reasoning.

**Strengths:**

1. The benchmark resembles real-world biological data engineering by sourcing its tasks from 150 published papers.
2. The paper presents a reasonable coverage of existing LLMs in the experiments and presents an error analysis of the agent failure modes.

**Weaknesses:**

1. The submission overlooks several related papers. For example, the annotation procedure and task styles are very similar to DiscoveryBench and ScienceAgentBench (both have biochem tasks), but neither is cited or discussed in the paper. Although I did not check each of the cited papers, the comparison in Table 1 to BixBench is incorrect: BixBench also involves multimodal data, and multiple-choice based evaluation is also objective. With these two items corrected, the differences between BixBench and BioDataLab are mostly erased. Thus, this benchmark’s novelty may be overstated.
2. If only 114 tasks are derived from 150 papers, it seems like the proposed annotation procedure may not be quite “cost-efficient” in terms of papers compared to other benchmarks, where multiple atomic tasks can be derived from the same paper. What is the reason for this low conversion rate in this benchmark, and why is it not a limitation?
3. What are the 22 evaluation functions? How do they ensure the evaluation objectiveness and correctness? What do the authors mean by “semantic parsers”, e.g., converting natural language to some formal language for evaluation? This part is not elaborated clearly in the paper.
4. What does each failure mode in the analysis mean? This part needs further clarifications.

**Questions:**

Please proof read the paper and fix typos, e.g. “354 tasks” (line 264).

**Details Of Ethics Concerns:**

All of the papers and resources used by this benchmark are not cited or attributed. This may hurt the original authors' intellectual properties and be subject to copyright and terms of use issues.

---

### Official Review · Reviewer_cFYj · 2025-11-01

**Soundness:** 2
**Presentation:** 3
**Contribution:** 3
**Rating:** 2
**Confidence:** 3

**Summary:**

This paper introduces BioDataLab, a benchmark for evaluating LLM-based agents on Biological Data Engineering (BDE) tasks. The authors formalize BDE as comprising four task categories: Database Querying, Tool Using, Custom Data Processing, and Scientific Reasoning. The benchmark consists of 114 real-world tasks curated from 150 peer-reviewed publications in Nucleic Acids Research, involving interactions with 50 biological databases and 56 bioinformatics tools. Each task includes manually generated ground-truth data and custom evaluation functions for complex scientific file formats. The authors evaluate state-of-the-art models (GPT-4.1, Claude 4, Gemini 2.5, DeepSeek V3.1, Qwen3) using the Biomni agent framework, finding modest overall success rates (best: 38.5%) and identifying four critical failure modes: brittle tool chaining, parameter hallucination, poor handling of scientific formats, and lack of long-horizon reasoning.

**Strengths:**

1. **A useful benchmark for biological data engineering tasks**: The benchmark addresses an important practical problem in biological research by focusing on data engineering workflows that are essential for constructing and processing biological datasets, which represents significant manual effort for researchers.
2. **Good methodological rigor**: The paper demonstrates strong methodological rigor through systematic quality control involving expert validation from biologists and bioinformaticians, multi-stage verification processes, and careful curation from peer-reviewed NAR database papers that ensures scientific authenticity.

**Weaknesses:**

1. **Overstated Novelty Claims**: The paper claims to be "the first comprehensive benchmark" for biological data engineering but fails to adequately acknowledge or differentiate from BixBench, and BaisBench, ScienceAgentBench, and other similar benchmarks, which cover highly similar ground with real-world biological data analysis tasks requiring multi-step reasoning and code generation. The distinction between "data engineering" versus "data analysis" is insufficiently articulated and may be too subtle to justify the "first" claim. The paper does not provide any direct comparison with existing benchmarks to demonstrate what unique aspects of biological data engineering are captured by their tasks. Without explicit task overlap analysis or cross-evaluation, it is unclear whether BioDataLab represents genuinely different capabilities or simply a parallel effort in the same space.
2. **Non-Orthogonal Task Categories**: The four task categories are not mutually exclusive, yet each task receives only one label. However, a task usually can involve multiple capabilities. The proposed categorization may lead to inaccurate failure analysis and capability assessment in the experiments. Intuitvely, each task should have multiple capability labels, and the failure analysis should be more fine-grained on the specific capability that an agent makes mistakes on.
3. **Limited Benchmark Scale**: The benchmark scale of 114 tasks is smaller than other benchmarks, making it difficult to claim comprehensiveness. The paper does not justify why this smaller scale is sufficient or explain what principle guided the selection to 114 tasks.
4. **Missing Critical Ablations.** The paper lacks critical ablation studies that would strengthen the technical contributions. There is no analysis of which specific evaluation functions are most discriminative, no comparison of step-level evaluation versus end-to-end evaluation for agent development, and no investigation of whether the domain-specific evaluators actually provide better signal than simpler metrics.
5. **Unaddressed Data Contamination.** The paper does not address potential data contamination concerns despite using tasks derived from public NAR database papers. There is no discussion of whether the training data for evaluated LLMs might have included these papers or their associated code repositories, which could artificially inflate performance estimates.

**Questions:**

1. What precisely distinguishes "data engineering" from "data analysis" in your framework? Can you provide concrete examples of tasks that exemplify pure data engineering versus those that would be classified as data analysis, and explain why this distinction matters for evaluating LLM agents?
2. How did you validate the correctness of your LLM-based multi-label classification judge? What is the agreement rate between GPT-4 judgments and human expert assessments? Did you measure inter-rater reliability for the cases where multiple evaluation approaches could apply?

---

### Official Review · Reviewer_KvGS · 2025-11-01

**Soundness:** 2
**Presentation:** 3
**Contribution:** 2
**Rating:** 2
**Confidence:** 3

**Summary:**

The paper introduces BIODATALAB, a benchmark for evaluating LLM-based agents on Biological Data Engineering (BDE) tasks. It curates 114 tasks from 150 NAR papers covering database querying, tool usage, data processing, and reasoning. The benchmark includes domain-specific evaluators for structured outputs and tests multiple LLMs under a unified agent framework.

**Strengths:**

1. Curating realistic, executable bioinformatics tasks addresses a clear gap between toy “tool use” benchmarks and real-world scientific data workflows.
2. The error taxonomy, covering tool-parameter hallucination, file-format handling, and multi-step reasoning failures, is informative for future work on scientific agents.

**Weaknesses:**

1. The paper reframes existing ideas (multi-tool LLM evaluation, structured output checking) within a biological context but introduces no new algorithmic or representational advances.
2. The 114 tasks heavily bias toward scripting and I/O operations from a single publication source (NAR). This does not capture the heterogeneity of real biological data engineering—ontology reconciliation, cross-schema integration, or experiment planning are absent.
3. There are no error analyses -- whether the failures stem from the agent wrapper, missing tool bindings, or brittle evaluation scripts rather than reasoning limitations.
4. The reproducibility is low --  no details on sampling parameters, compute infra, exact lists of papers and tasks, agent framework.

**Questions:**

See weaknesses above

---

### Meta-Review · Area_Chair_iCzQ · 2026-01-02

**Summary:**

All reviewers agree that the paper introduces BioDataLab, a benchmark targeting biological data engineering tasks derived from real scientific workflows. It curates totally 114 tasks from 150 NAR papers covering database querying, tool usage, data processing, and reasoning. The benchmark includes domain-specific evaluators for structured outputs and tests multiple LLMs under a unified agent framework.

However, a majority of reviewers raise serious concerns about overstated novelty, limited benchmark scale and diversity, insufficient differentiation from existing benchmarks, and missing analyses or ablations that would substantiate the claimed contributions. Concerns are also raised regarding evaluation design transparency, task categorization validity, reproducibility, and potential ethics or copyright issues. As a result, most reviewers recommend rejection, with only one reviewer expressing marginal support.

**Reviewer Concerns:**

The authors have not submitted their rebuttal.

**Reviewer Scores:**

Reviewer KvGS: 2

Reviewer cFYj: 2

Reviewer iMuJ: 2

Reviewer Beei: 6

---

### Decision · Program_Chairs · 2026-01-26

Reject